# MARVEL: Multidimensional Abstraction and Reasoning through Visual Evaluation and Learning

**Yifan Jiang**[1*]    **Jiarui Zhang**[1*]    **Kexuan Sun**[1*]    **Zhivar Sourati**[1]
**Kian Ahrabian**[1]    **Kaixin Ma**[2]    **Filip Ilievski**[3]    **Jay Pujara**[1]

[1]Information Sciences Institute, University of Southern California
[2]Tencent AI Lab, Bellevue, WA
[3]Department of Computer Science, Vrije Universiteit Amsterdam

`{yjiang44,jzhang37,kexuansu,souratih,ahrabian}@usc.edu`
`kaixinma@global.tencent.com`, `f.ilievski@vu.nl`, `jpujara@isi.edu`

## Abstract

While multi-modal large language models (MLLMs) have shown significant progress across popular visual reasoning benchmarks, whether they possess abstract visual reasoning abilities remains an open question. Similar to the Sudoku puzzles, abstract visual reasoning (AVR) problems require finding *high-level patterns* (e.g., repetition constraints on numbers) that control the *input shapes* (e.g., digits) in a specific *task configuration* (e.g., matrix). However, existing AVR benchmarks only consider a limited set of patterns (addition, conjunction), input shapes (rectangle, square), and task configurations ($3 \times 3$ matrices). And they fail to capture all abstract reasoning patterns in human cognition necessary for addressing real-world tasks, such as geometric properties and object boundary understanding in real-world navigation. To evaluate MLLMs' AVR abilities systematically, we introduce **MARVEL** founded on the core knowledge system in human cognition, a multi-dimensional AVR benchmark with 770 puzzles composed of six core knowledge patterns, geometric and abstract shapes, and five different task configurations. To inspect whether the model performance is grounded in perception or reasoning, MARVEL complements the standard AVR question with *perception questions* in a hierarchical evaluation framework. We conduct comprehensive experiments on MARVEL with ten representative MLLMs in zero-shot and few-shot settings. Our experiments reveal that all MLLMs show near-random performance on MARVEL, with significant performance gaps (40%) compared to humans across all patterns and task configurations. Further analysis of perception questions reveals that MLLMs struggle to comprehend the visual features (near-random performance). Although closed-source MLLMs, such as GPT-4V, show a promising understanding of reasoning patterns (on par with humans) after adding textual descriptions, this advantage is hindered by their weak perception abilities. We release our entire code and dataset at `https://github.com/1171-jpg/MARVEL_AVR`.

## 1 Introduction

Recent advances in novel training pipelines, computational resources, and data sources have enabled multi-modal large language models (MLLMs) [21, 49, 37, 16, 8, 10] to show strong visual reasoning ability in tasks that require combining both visual and textual cues [64], such as visual question answering [7, 22] and visual commonsense reasoning [70, 76]. These tasks are typically under real-world settings [40]. On the other hand, abstract visual reasoning (AVR) [27, 77] focuses on

---

* Authors contributed equally

38th Conference on Neural Information Processing Systems (NeurIPS 2024) Track on Datasets and Benchmarks.

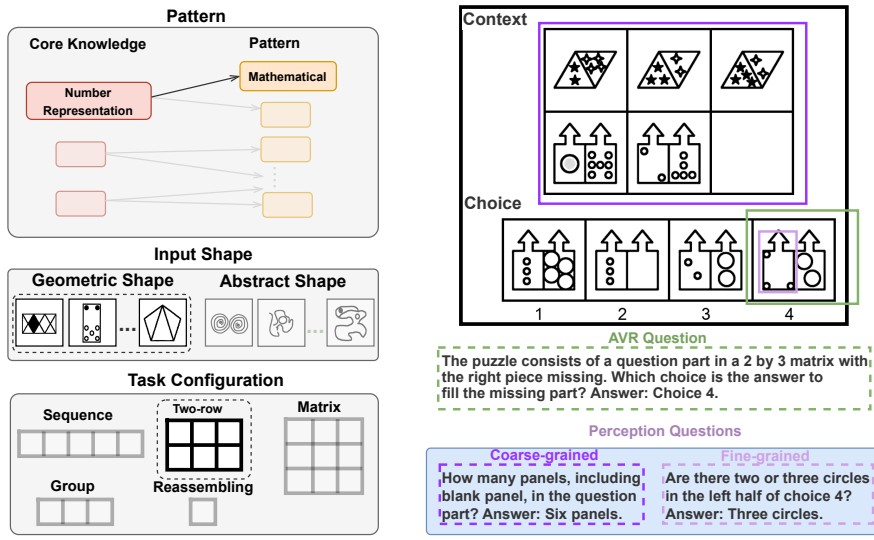

Figure 1: An abstract visual reasoning puzzle in **MARVEL**. The puzzle contains **mathematical** pattern governing the element number in **geometric shapes** with **two-row** task configuration. The AVR question focuses on the final answer for the puzzle, while the perception questions focus on the fine-grained detail about one choice or coarse-grained detail over the whole puzzle. In the example, the left-side elements (black stars/circles) increase by one in each panel, while the right-side elements (white stars/circles) in the first panel equal the sum of those in the second and third panels.

recognizing patterns among 2D shapes and their attributes. As the puzzle shown in Figure 1 (top-right), AVR problems require identifying the hidden pattern (addition and subtraction) that governs the input shapes and their attribute (number of stars/circles) in a task configuration ($2 \times 3$ matrix). AVR ability is crucial for various practical applications, including human pose estimation (understanding pose through abstract representation) [50], and anomaly detection(finding outliers in videos) [55]. AVR ability is also indispensable for developing artificial general intelligence (AGI) [40]. This significance encourages fundamental research on evaluating MLLMs against AVR benchmarks [4, 73].

However, two significant issues remain unsolved in evaluating MLLMs comprehensively. 1) **The scope of current AVR benchmarks is not fully cognitively supported and fails to encompass the variety of real-world scenarios**[63]. Some datasets, such as RAVEN [77], only cover a few reasoning patterns (mostly in mathematical patterns) over a limited set of input shapes arranged in a predetermined configuration of puzzle panels. Human cognition builds the foundation for inference and skill acquisition in the real world[59]. Without theoretical foundations, existing benchmarks lack diverse reasoning patterns in real-world tasks and fail to provide a holistic evaluation of MLLM's ability[15], which can not be solved by simple joint evaluations[40]. 2) **Most prior studies** [11, 28] **employ an end-to-end evaluation framework, leaving it unclear whether the model's performance is attributed to perception or reasoning**[56]. Current visual literature claims they are both compositional components of the visual reasoning process and should be treated separately for in-depth analysis[4, 60, 75].

To address the limitation of evaluation scope (issue 1), we introduce **MARVEL**, a multi-dimensional abstract visual reasoning benchmark designed to evaluate MLLMs across six patterns, both geometric and abstract shapes and five task configurations. To ensure cognitive foundations and real-world applicability, MARVEL's underlying reasoning patterns are rooted in key core knowledge of human cognition, observed in newborn infants, necessary for reasoning about their environment (real-world scenario) [59] even without a clear real-world understanding (abstraction). We crawl relevant puzzles from publicly available websites, manually filter low-quality and irrelevant puzzles based on the expanded patterns and input shapes, and reformat them into different task configurations. We annotate the AVR question by briefly describing the puzzle and asking for its answer. In total, we collect 770 diverse and high-quality puzzles assessing abstract visual reasoning abilities (Figure 1).

| Dimension | | RAVEN | G-set | VAP | Bongard-LOGO | SVRT | DOPT | ARC* | **MARVEL** |
|---|---|---|---|---|---|---|---|---|---|
| **Input Shape** | Geometric | ✔ | ✔ | ✔ | | | ✔ | ✔ | ✔ |
| | Abstract | | | | ✔ | ✔ | | | ✔ |
| **Pattern** | Temporal Movement | ✔ | ✔ | | | | ✔ | ✔ | ✔ |
| | Spatial Relationship | | | | | ✔ | ✔ | ✔ | ✔ |
| | Quantities | ✔ | ✔ | ✔ | ✔ | ✔ | ✔ | ✔ | ✔ |
| | Mathematical | ✔ | ✔ | ✔ | | | | ✔ | ✔ |
| | 2D-Geometry | | | | ✔ | ✔ | | | ✔ |
| | 3D-Geometry | | | | | | | | ✔ |
| **Configuration** | Sequence | | | ✔ | | | ✔ | | ✔ |
| | Two-row | | | | | | | | ✔ |
| | Matrix | ✔ | ✔ | | | | | | ✔ |
| | Group | | | | ✔ | ✔ | | | ✔ |
| | Reassembling | | | | | | | | ✔ |
| Perception Question | | | | | | | | | ✔ |

Table 1: Comparing MARVEL to related benchmarks: RAVEN [77], G-set [41], VAP [27], Bongard-LOGO [47], SVRT [19], ARC [15], DOPT [67]. *ARC puzzles are provided in a generative format.

To determine if the model's performance is based on perception or reasoning (issue 2), we also provide a hierarchical evaluation framework by enriching each puzzle with perception questions focusing on perceiving puzzles' visual details (e.g., number of grids, edges of a triangle) to measure models' reasoning consistency [31, 56]. We conduct comprehensive experiments on MARVEL involving different model structures, model sizes, and prompting strategies. Our experiments reveal that all MLLMs show near-random performance in all patterns, even with few-shot demonstrations and prompt engineering, leaving a huge gap (40%) in the abstract reasoning ability of humans. An in-depth analysis based on perception questions points out that MLLMs' performance is hindered by their fine-grained visual feature comprehension, failing to provide foundations for subsequent abstract reasoning. Our contributions can be summarized as follows: 1) **A novel multidimensional AVR benchmark**, **MARVEL**, which consists of six patterns rooted in cognitive theory across five distinct task configurations. 2) **A hierarchical evaluation framework** incorporating perception questions with AVR questions to enable fine-grained diagnosis of model capability. 3) **Extensive experiments** on a wide range of state-of-the-art MLLMs with various prompting strategies, providing insights into the connection between their perception and reasoning abilities.

## 2 Related Work

**MLLM Evaluations.** Benefiting from the rich representation from visual encoders [52] and strong reasoning ability of LLMs [62, 14], MLLMs [33, 16, 48, 37] have been applied to solve not only traditional vision-language tasks, such as image captioning [1, 74], visual question answering [23, 43, 29, 57] and refer expression comprehension [32, 24], but also on more complicated scenarios, such visually-grounded conversation [37, 5], multimodal web/UI agents [25, 81, 72] and embodied tasks [17]. Besides end-to-end evaluation, several recent works also try to reveal MLLMs' visual shortcomings from different aspects, including visual details [79, 69], perceptual bias [78], and small visual pattern recognition [61]. Although some of the existing benchmarks have accessed MLLM's mathematical visual reasoning abilities requiring an understanding of abstract and geometry shapes [38, 39], their evaluation still heavily relies on textual descriptions. In contrast, AVR benchmarks assess MLLMs' ability under diverse patterns with only visual understanding settings.

**AVR Benchmarks.** AVR problems have great potential impact on various domains [40, 51, 55], sparking interests in evaluating MLLMs on AVR benchmarks [4, 46, 45]. Existing AVR benchmarks present the evaluation in a wide range of formats, such as selective completion [77, 28, 11, 67], group discrimination [19, 47] and generative completion [15]. However, less attention is paid to the scope and pattern of the AVR benchmark; most focus only on a few simple abstract patterns and testing models end-to-end without considering the intermediate perception and reasoning procedures [46, 44]. In contrast, MARVEL not only includes geometric and abstract shapes and five different task configurations but also builds its reasoning patterns based on the core knowledge system in cognitive science, enriching abstract reasoning that exists in the real world. Inspired by prior analysis of visual details and perceptual bias, MARVEL introduces perception questions to ensure the MLLMs

correctly perceive the presented visual patterns. MARVEL and related AVR benchmarks are compared in Table 1.

## 3 MARVEL Benchmark Construction

As a multidimensional benchmark for AVR, MARVEL covers different task configurations (Section 3.1), various input shapes (Section 3.2), as well as different reasoning patterns involved in the puzzles (Section 3.3). We present the data collection process in Section 3.4.

### 3.1 Task Definition and Configurations

Each puzzle in MARVEL consists of a context on the top and possible choices ($c_i; i \in \{1, 2, 3, 4\}$) to choose from at the bottom, formatted in a multiple-choice question answering setting (see Figure 1 for an example). The context part consists of $n$ puzzle panels ($p_1, p_2, \ldots, p_n, p_b$ with $p_b$ being a blank panel), with their specific number and arrangement driven by a task configuration and reasoning pattern, $P$, that governs the relationship between puzzle panels. The choice will be considered the correct answer and fill in $p_b$ that can satisfy the following equation: $P(p_1, p_2, \ldots, p_n) = P(p_1, p_2, \ldots, p_n, c_c)$.

Puzzle panels in MARVEL are organized in five task configurations. We visualize these configurations in Figure 1 (see detailed examples in Appendix D).

1. **Sequence Format** arranges panels in a $1 \times n$ line ($n \in [4, 7]$).
2. **Two-row Format** presents panels in a $2 \times 3$ matrix*, $p_1^1, p_2^1, p_3^1$ and $p_1^2, p_2^2, p_3^2$. The solution requires identifying the same pattern at the first row and the second row, which is $P(p_1^1, p_2^1, p_3^1) = P(p_1^2, p_2^2, c_c)$.
3. **Matrix Format** organizes panels in a 3 by 3 matrix, the pattern can be reflected in either row- or column-wise way: $P(p_1^1, p_2^1, p_3^1) = P(p_1^2, p_2^2, p_3^2) = P(p_1^3, p_2^3, c_c)$ or $P(p_1^1, p_1^2, p_1^3) = P(p_2^1, p_2^2, p_3^2) = P(p_3^1, p_3^2, c_c)$.
4. **Group Format** has three panels in the question part ($p_1, p_2, p_b$) with one choice reflecting the context's pattern and other choices differing: $P(p_1, p_2, c_c) \neq P'(choices - c_c)$.
5. **Reassembling Format** is designed for *3D-Geometry* pattern visualizes 3D geometric shapes, such as cubes or tetrahedrons, in the choice section of the puzzle. The context section provides a panel featuring the unfolded 2D diagram of one of the choices. Models must reason about the visual details in the 2D diagram to identify the correct 3D shape.

### 3.2 Input Shapes

As shown in Figure 1, each panel of a puzzle contains various shapes that can be generally differentiated into two types [40]:

1. **Geometric Shapes** are easily described and come from a limited vocabulary. For instance, a square is a shape that has four sides of equal length and four equal angles. Most existing AVR benchmarks [77, 27] focus on elementary shapes such as oval, rectangle, triangle, and trapezoid. MARVEL includes geometric shapes consisting of more than two different elementary geometric shapes to mitigate the issue and improve the complexity.
2. **Abstract Shapes** come from a wide set of possibilities and vary widely from one problem to another[40]. Unlike geometric shapes, which are typically fixed, easily describable, and belong to a finite set (e.g., both a larger square and a black square are classified as squares due to shared fundamental properties), abstract shapes are atypical, lacking fixed properties and marked by variability and complexity. They provide a fair step as most MLLMs encounter the shapes for the first time and are gaining more preference for AVR-related research [19, 47].

### 3.3 Core Knowledge and Patterns

Core knowledge theory [59] from cognition science is largely shared among humans and particularly for human infants. Human infants with no real-world knowledge and limited experience represent

---

*$p_a^b$: the panel on the $a$ th row and $b$ th column of matrix.

their environment using abstraction patterns. These abstraction patterns can be categorized into four types of core knowledge, which is the foundation for inference and reasoning [34] in real-world scenarios. We do not feature the agent representation core knowledge because it concentrates on goal-directed and interactive action, which is not adaptable in MARVEL setting. For the other three types of core knowledge, we expand each into two patterns for a fine-grained assessment of abstract reasoning in MARVEL, based on insights drawn from contemporary cognitive literature:

1. **Object Core Knowledge** represents objects' spatiotemporal motions and their contact, enabling humans to predict objects' movement and perceive object boundaries. We expanded this core knowledge to *Temporal Movement Pattern* focusing on the related position change or movement [58] and *Spatial Relationship Pattern* examining objects' relative positional relationship [2].
2. **Number Core Knowledge** helps infants process abstract representations of small numbers and perform comparisons. We include *Quantities Pattern* testing the accuracy of number comprehension [71] and *Mathmatical Pattern* for elementary mathematical operations [9].
3. **Geometry Core Knowledge** captures the environment's geometry, which helps humans orient themselves in their surroundings. We divide the concept into *2D-Geometry Pattern* [26] and *3D-Geometry Pattern* [13].

### 3.4   Data Collection

We collect puzzles from several public resources websites[1] and filter out unfit or low-quality data by three human annotators based on the puzzle's input shapes (some puzzles contain textual information) and patterns. Unaligned puzzles are first segmented into panels and then reassembled into the correct task configuration. To ensure each pattern in each task configuration has at least 45 puzzles[2], we also manually created 220 puzzles by following the pattern in existing data and replacing the input shape drawn from scratch. Each puzzle contains an AVR question (Figure 1) generated from templates based on their task configuration. AVR questions provide a brief description and ask only for the puzzle's final answer, which is widely adopted in previous AVR benchmark [40]. In the end, MARVEL includes 770 high-quality puzzles over six high-level patterns across five distinct task configurations. In Table 1, we compare MARVEL with existing AVR benchmarks to show its comprehensive scope.

## 4   Hierarchical Evaluation Framework

Previous works evaluate MLLMs on AVR benchmarks with end-to-end setting only [46, 45], potentially overlooking shortcut learning and inductive biases [40]. On the other hand, precisely comprehending visual details is the foundation for subsequent reasoning in AVR problems [20]. We enrich MARVEL puzzles with perception questions [56] designed to test models' perception ability on visual details (Figure 1). We design a hierarchical evaluation framework by combining two types of perception questions with AVR questions (Figure 1) to examine if model accuracy is based on perception and reasoning. For each puzzle, our framework provides three coarse-grained questions and one pattern-related fine-grained question:

**Coarse-grained Perception Question** in an open-ended fashion aims to test if models can understand the task configuration correctly by directly asking about the number of panels in puzzles. We use templates to generate three questions focusing on the number of panels in the context part, choice part, and the whole puzzle. We remove the choice index (the number marking each choice panel) when testing models with this question to avoid shortcut learning.

**Fine-grained Perception Question** in binary-choice format examines models' understanding of input shapes, which focus on the visual details categorized by Tong et al. [60] such as shape attributes (number of edges) and spatial relationship (left, right) based on the pattern contained in the puzzle. For example, in Figure 1, the fine-grained perception question tests whether models can understand the number of circles because the puzzle is based on *Mathematical Pattern*. For each puzzle, we randomly pick one choice panel in the puzzle and manually create questions with two

---

[1] https://www.gwy.com/; https://www.chinagwy.org/
[2] Some patterns can not be presented in specific configurations. For example, the *Mathmatical Pattern* can not be adapted to group format as it requires comparison between adjacent panels.

choices. The correct answer is randomly placed to avoid inductive bias[3]. We have five types of questions based on the pattern and how it adapts to the input shape, which are listed with examples:

1. **Location**: Is the dot outside or inside of the star in choice 4?
2. **Color**: Is the triangle black or white in choice 1?
3. **Shape**: Is there a circle or a triangle inside choice 3?
4. **Quantity**: Are there five or four circles in choice 2?
5. **Comparison**: Are the left and right halves of the rectangle in choice 3 the same?

## 5 Experimental Setup

**Closed-source MLLMs.** We include API-based MLLMs including 1) GPT-4 [48], 2) Gemini [21] and 3) Claude3 [6]. With the massive computation and training data, these models show promising performance on a wide range of visual-focused tasks [23, 70]. We evaluate closed-source MLLMs in both zero-shot and few-shot [12] settings.

**Open-source MLLMs.** We include MLLMs smaller than 13B due to our limited computing resources: 1) InstructBLIP [16], 2) BLIP-2 [33], 3) Fuyu [10], 4) Qwen-VL [8] and 5) LLaVA [36]. We only evaluate these MLLMs in a zero-shot setting due to their single-image input settings [80].

**Human Evaluation.** To access the upper bound performance on MARVEL, we simulate a realistic human assessment by inviting 30 annotators aged from 10 to 50 years to solve a subset of MARVEL and ensure each subset contains every pattern in all task configurations. We compute the average performance of these 30 annotators as the human baseline. Each puzzle is solved by at least two annotators. We invited three annotators to solve perception questions and report their average performance. The demonstrations and instructions used are presented in Appendix J.

**Evaluation Metrics.** Following a similar setting as previous research evaluating MLLMs on AVR benchmarks [4], we use regex matching to extract the choices picked (e.g., "choice 4" in the response "The correct answer is choice 4."), with failure cases re-extracted by GPT-4 [3, 82]. We use accuracy as the metric, commonly used for evaluating multiple-choice questions, and has been utilized by many AVR papers [27, 77]. Based on the hierarchical evaluation framework, we evaluate MLLMs with two types of accuracy-based metrics:

1. **Instance-based Accuracy** considers questions separately. We report accuracy results for *AVR question* and *fine-grained perception question*.
2. **Group-based Accuracy** considers questions as groups to assess the consistency in model reasoning [31, 75]. The model receives a score of 1 only if it correctly answers all questions within the same group. We report the group-based accuracy result of combining all three coarse-grained perception questions and the further result after introducing fine-grained and AVR questions into the group.

## 6 Results

We focus on five research questions: *1) What's the abstract reasoning ability on visual puzzles of current SOTA MLLMs? 2) Can MLLMs do better with different few-shot prompting strategies? 3) How do MLLMs perform on different patterns and task configurations? 4) To what extent do MLLMs visually understand the puzzle? 5) Do they show consistent reasoning ability?*

**Overall Performance.** The AVR question results are shown in Table 2. Human performance reaches 68.86%, with a standard deviation of 9.74, confirming the validity and challenging nature of MARVEL. For both open and closed source categories, **all models show near-random performance with a huge gap (40%) compared to human performance**, in which closed-source MLLMs (avg: 25.7%) perform slightly better than open-sourced ones (avg: 24.0%). We observed an extremely imbalanced distribution in the outputs of some MLLMs. For example, BLIP-2 consistently selecting choice 1 for all puzzles (marked † in Table 2). We tried different approaches with our best effort to avoid potential bad prompts or engineering settings, including adding question marks in the black panel, replacing the choice index with letter (1 → A), and changing the description in the AVR

---

[3]The inductive bias (also known as learning bias) of a learning algorithm is the set of assumptions that the learner uses to predict outputs of given inputs that it has not encountered. For example, the model may always output 'D' for unfamiliar questions

Table 2: Main zero-shot accuracy over MARVEL across all MLLMs in two accuracy metrics: $Prec^C$ = group-based accuracy over all coarse-grained perception questions (model must answer all three questions correctly), $Perc^{C\&F}$ = group-based accuracy combining all perception questions (coarse/fine-grained), AVR = AVR Question. The best performance among all models is in **bold**, and the best result in two MLLMs categories is underlined. *Gemini refuses to answer the puzzle due to safety problems in 7% cases so the performance is computed based on the left set. † notes the result is attributed to inductive bias.

| Category | Model | AVR Question | Fine-grained Perception | Perc$^C$ | Perc$^{C\&F}$ | Perc$^{C\&F}$ & AVR |
|---|---|---|---|---|---|---|
| | **Random** | 25.00 | 50.00 | - | - | - |
| **Open-source MLLMs** | Qwen-VL (7B) | 19.61 | 37.27 | 0.52 | 0.39 | 0.00 |
| | Fuyu (8B) | 24.29† | 34.94 | 0.00 | 0.00 | 0.00 |
| | BLIP-2 (FlanT5$_{XXL}$-11B) | 24.81† | 53.38 | 1.04 | 0.52 | 0.26 |
| | InstructBLIP (Vicuna-13B) | 24.68† | 49.48 | 0.00 | 0.00 | 0.00 |
| | LLaVA-1.5 (Vicuna-13B) | 26.36 | 51.43 | 1.14 | 0.52 | 0.13 |
| **Closed-source MLLMs** | GPT-4V | 22.34 | 51.56 | 18.31 | 9.22 | 2.73 |
| | GPT-4o | 27.79 | **68.44** | 62.73 | 43.77 | **12.21** |
| | Gemini-pro-vision* | 25.06† | 44.42 | 15.19 | 6.75 | 1.69 |
| | Claude3 (Sonnet) | 26.49† | 50.91 | 38.70 | 19.87 | 5.06 |
| | Claude3 (Opus) | **28.83** | 47.27 | 44.94 | 20.13 | 5.97 |
| | **Human** | 68.86 ± 9.74 | 98.67 | - | - | - |

question. None of them can mitigate and may even exacerbate the issue, highlighting the potential inductive biases [66] in models. Among open-source MLLMs, LLAVA performs the best, yet the gap is very small, and it is unclear whether the gain comes from its larger model size. In closed-source models, even the strongest MLLMs, Claude3 (Opus) and GPT-4o, which demonstrated promising results on various vision tasks [6], failed to present a significant performance difference from the random baseline. Claude3 (Sonnet) and Gemini also have imbalanced output distributions, with both selecting choice 4 in most cases.

**Impact of Few-Shot CoT.** Given the poor zero-shot performance and the models' capability of in-context learning [12], we explore few-shot prompting with Chain-of-Thought (CoT) [68] to guide MLLMs with abstract reasoning patterns. We experiment with all closed-source MLLMs in one-shot and two-shot settings using manually created CoT context, similar to Yang et al. [73]. For each puzzle, we randomly select puzzles with the same pattern in the sequence task configuration and annotate the CoT reasoning with answers as demonstrations. We chose the sequence task because it is more straightforward (only along the sequence) than other configurations. Each demonstration is formatted as image-text pairs. Our result is shown in Figure 2, and we present the full results in Appendix I. The few-shot demonstrations show a marginal positive impact on GPT-4V and a decreasing trend on Claude3 (Opus).

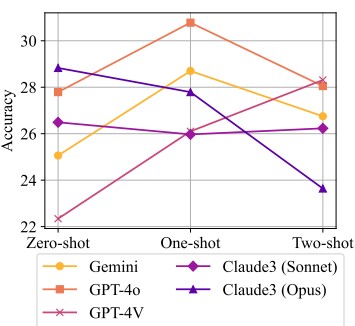

Figure 2: MLLMs performance in different few-shot COT.

Further analysis reveals that the main improvement in GPT-4V's results lies in the *3D-Geometry pattern*. As this pattern focuses on reassembling, the demonstration can guide the model to pay attention to the relative position of each side of the object. However, since most patterns are uniquely implemented on different input shapes and their attributes, the model struggles to learn generalizable patterns from the few-shot demonstrations. Figure 3 provides an example of zero-and few-shot results from Claude3 (Opus). With the demonstration, the model learns to focus on the correct pattern (blue) at the beginning of the reasoning. However, it fails to adapt precisely to the input shapes in the puzzle (red), leading to errors in subsequent reasoning. We also test different prompt engineering approaches, including selecting demonstration samples from different 1) patterns, 2) task configurations, and 3) prompting MLLMs by dividing puzzles panel by panel. None of these approaches yields a positive impact; instead, they lead to a significant drop in performance (Appendix I). Given the complexity

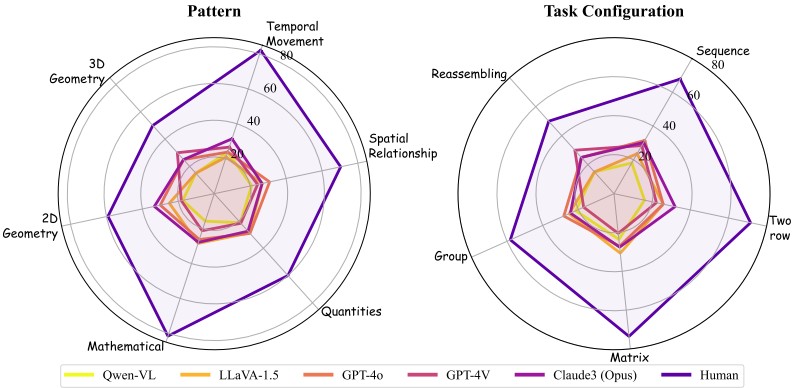

Figure 4: MLLMs and human performance across patterns and task configurations.

and challenging nature of the dataset, the effectiveness of few-shot prompting on MARVEL remains minimal. The inductive bias is mitigated in few-shot settings.

**Performance on Different Patterns and Configurations.** We further break down the results[4] into different patterns and task configurations in Figure 4 (full results in Appendix E). In general, MLLMs show near-random performance on all patterns and task configurations. Among the six patterns, *3D-Geomertry* pattern is the most challenging for humans and MLLMs. The difficulty may be rooted in the requirement for the 3D imaginative ability [42] and the relatively lower frequency of such data in the models' pre-training datasets, which leads to a significant gap (12.5%) between open and closed-sources MLLMs. On the other hand, *2D-Geomertry*, involving understanding geometric attribute of input shape, seems relatively easier for MLLMs, evidenced further by their proficiency on similar tasks such as interpreting graphs plots [73].

Among the five task configurations, the two-row and matrix formats provide more panels (i.e., information) to verify the pattern, facilitating puzzle-solving. On the contrary, the group and reassembling formats containing the least panel number tend to be challenging. Four out of five MLLMs rank 1st in different task configurations, which verifies our assumption of potential bias in single-configuration evaluation. According to their pre-training dataset, models may be familiar with specific input types, highlighting the necessity of MARVEL's multidimensional settings.

**Perception Ability on Visual Details.** Visual perception forms the foundation for advanced reasoning [54]. By incorporating perception questions, our

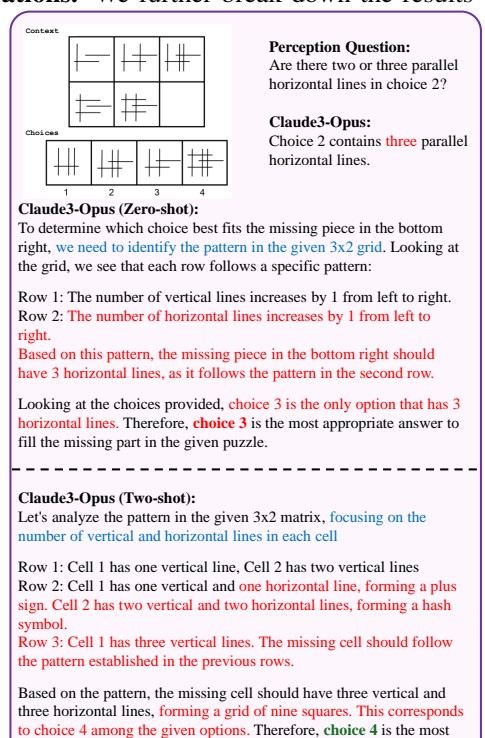

Figure 3: Perception question, zero- and two-shot example of Claude3 (Opus).

hierarchical evaluation framework effectively investigates to what extent the models understand the visual information from the puzzle. In Table 2, closed-source MLLMs demonstrate more robust performance on coarse-grained perception group accuracy compared to open-sourced MLLMs, with a performance gap ranging from 14.05% to 61.59%. However, even the best model fails to reach an acceptable accuracy (<65%), indicating that current MLLMs struggle to simultaneously understand the number of grids, choices, and the puzzle as a whole, despite their promising performance on real-world datasets [30]. The simplicity of the coarse-grained perception questions (all puzzles contain less than 13 panels) highlights the poor perception ability of current MLLMs in the abstract visual reasoning domain. Fine-grained perception questions further confirm this argument, with

---

[4]We only select MLLMs that do not show significantly imbalanced output distributions.

Table 3: Performance of different models after introducing text description in the input. † notes the result is attributed to inductive bias.

| Input | Open-sourced MLLMs | | | | | Closed-sourced MLLMs | | | | |
|---|---|---|---|---|---|---|---|---|---|---|
| | Qwen-VL | Fuyu | Blip-2 | InstructBLIP | LLaVA-1.5 | GPT-4V | GPT-4o | Gemini | Claude3 (Sonnet) | Claude3 (Opus) |
| AVR | 23.16 | 23.16† | 23.16† | 23.16† | 21.05 | 21.05 | 23.16 | 26.32† | 27.37† | 30.53 |
| AVR+Text | $24.21_{\uparrow 1.05}$ | $26.32_{\uparrow 3.16}$ | $26.32_{\uparrow 3.16}$ | $13.68_{\downarrow 9.48}$ | $28.42_{\uparrow 7.37}$ | $65.26_{\uparrow 44.21}$ | $58.95_{\uparrow 35.79}$ | $37.89_{\uparrow 11.57}$ | $49.47_{\uparrow 22.10}$ | $55.79_{\uparrow 25.26}$ |

all models except GPT-4o showing near-random performance. Further analysis of fine-grained perception performance based on five categories (Table in Appendix F) reveals that models perform relatively better at color perception but have difficulty recognizing location (e.g., 'a' is on the left of 'b'). We hypothesize that the difficulty in understanding location stems from the lack of labeled data on location and relations during training, especially in abstract visual understanding. In contrast, the models' color perception is well-trained during their multi-modal alignment, and the simplicity of RGB understanding allows for easier transfer to the abstract domain.

**Consistency of Model Reasoning.** The further group-based accuracy ($Prec^{C\&F}$ and $Prec^{C\&F}\&AVR$) shows that **no model can solve the AVR puzzles with consistent reasoning**, with the best model reaching only 12.21% group accuracy. Based on the result of our evaluation framework, we hypothesize the inconsistency stems from their poor visual perception ability [20]. As shown in Figure 3, the model's reasoning is based on the perception of the puzzle (e.g., number of lines), which needs to be completely precise to support correct reasoning. The perception questions in our framework reveal that the model cannot clearly understand the number of lines, explaining why it fails to answer the puzzle even with correct hints (few-shot). A single error in visual feature perception can impact reasoning since the correct pattern must apply to all puzzle shapes. The densely packed information distribution—where the majority of the puzzle remains blank—ensures that each piece of visual perception is an essential foundation for subsequent reasoning. That also explains why GPT-4o struggles to achieve significant reasoning performance even with the highest performance on fine-grained perception questions. However, the importance of visual detail perception has received little attention in previous evaluations [46, 45], highlighting the significance of our new evaluation framework.

## 7 A Lens to Reasoning Through the Haze of Perception

Since poor visual perception is the main obstacle to improving MLLMs' abstract reasoning ability, in this section, we conduct three additional experiments to understand these models' potential when perceptual barriers are mitigated (full detail in Appendices G and H).

In the first experiment, we analyze the model's perception ability on the question part by presenting models with the same puzzle but asking for possible underlying patterns in a multiple-choice setting instead of the whole AVR reasoning question. As shown in Figure 5, closed-source models show non-random results when reasoning about the underlying pattern, while nearly all open-source models struggle to outperform random baselines. This gap indicates that closed-source models partially understand the patterns, but more accurate visual perception is needed to complete the entire task.

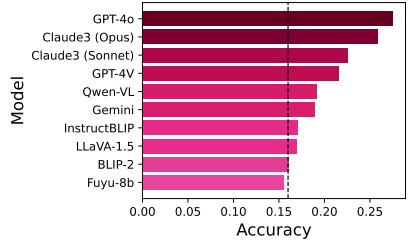

Figure 5: MLLMs performance on pattern classification. The black dotted line represents the random baseline.

To further alleviate perception barriers, we add accurate text descriptions of the puzzle on a subset of MARVEL (Table 3). The result shows a significant boost in performance, with GPT-4V achieving human-level accuracy (65%). On the contrary, open-source MLLMs still lag behind, indicating that while enhanced textual descriptions can improve performance, they do not fully bridge the gap between closed-source and open-source models. This further supports that closed-source models possess superior reasoning capabilities often overshadowed by visual perception. The distinction between different MLLMs also underscores the potential effectiveness of MARVEL in evaluating AVR ability, particularly when the weakness of visual perception is addressed.

To address concerns about domain shifts affecting model performance, we conducted domain adaptation experiments to evaluate the effectiveness of fine-tuning on perceptual tasks. We fine-tuned

LLaVA-1.5 [35] using Qwen-1.5B [8], with a 4:1 training-validation split over 10 epochs. The model was trained on both AVR and perception questions, and performance was measured on the validation set after each epoch. For AVR questions, the average accuracy was 19.76%, ranging from 12.99% to 29.87% (random chance 25%). For perception questions, average accuracy reached 59.74%, ranging from 53.89% to 62.98% (random chance 50%). These results indicate that poor performance is not solely attributable to domain shifts, aligning with prior studies [38, 53], which emphasize the inherent complexity of perception tasks that may require multi-task integration. The perception questions in MARVEL thus serve as a valuable benchmark for assessing MLLMs' perceptual capabilities.

## 8 Conclusion

In this work, we develop MARVEL, a multidimensional abstract visual reasoning benchmark consisting of 770 puzzles with both geometric and abstract input shapes across six patterns and five task configurations. We also design a hierarchical evaluation framework that enriches MARVEL with perception questions to enable granular analysis of models' visual details understanding and reasoning consistency. Our comprehensive experiments with ten SOTA MLLMs reveal a huge gap in abstract visual reasoning ability between (40%) humans and MLLMs, where all MLLMs often perform close to random. Further analysis based on our evaluation framework shows MLLMs' poor perception ability in understanding visual details, which hinders their subsequent reasoning and leads to poor AVR performance. We hope future works can build on the foundation of MARVEL for enhancing MLLM abstract visual perception and reasoning abilities.

## Acknowledgements

We appreciate Fred Morstatter for very helpful comments. We thank Tian Jin and Jiachi Liang for their assistance in data collection. This research was sponsored by the Defense Advanced Research Projects Agency via Contract HR00112390061.

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

## A Limitation

While our work draws from cognitive science to address gaps in current AVR literature, it's important to note that research in human cognition is ongoing. The exact nature of innate human core knowledge remains an open question, and it is necessary to continually update the scope of MARVEL as related cognitive studies progress.

Also, We chose a multiple-choice QA format with additional perception questions for hierarchical evaluation. Future research should explore better evaluation metrics such as open-ended generation or interactive (multi-step) setups, providing the possibilities to implement the fourth core knowledge: agent representation related to goal-directed and interactive actions.

## B Ethical Considerations

As our abstract visual reasoning puzzles are published on various websites, checking all original licenses comprehensively is challenging. However, the website owners permit printing and downloading for non-commercial use without modification. We will require future dataset users to sign a document stating that the data will be used solely for research purposes before providing access.

We also emphasize that puzzles in MARVEL are intended for research use only and should not be used to make critical decisions about individuals' capabilities. Such misuse could cause undue pressure and anxiety for participants and may not accurately reflect their true potential or abilities in real-world scenarios.

Puzzles in MARVEL only contain geometric and abstract shapes, which leaves no space for the disclosure of any personally identifiable information or offensive content. We still go through all the puzzles to make sure that no harmful or personal information exists in MARVEL.

## C Data Source

Our code and data are available at `https://github.com/1171-jpg/MARVEL_AVR/tree/main`. and the link to our dataset website is `https://marvel770.github.io/`

## D Data Examples

We present examples with 5 different task configurations and 6 patterns in Figures 6 to 12.

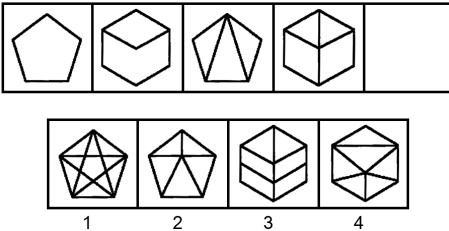

Figure 6: The example is formatted in *Sequence* configuration with the *Quantities* pattern. The answer to this puzzle is B.

## E Performance on Different Pattern and Tasks

Table 4 and Table 5 show MLLMs' AVR question performance on different patterns and tasks.

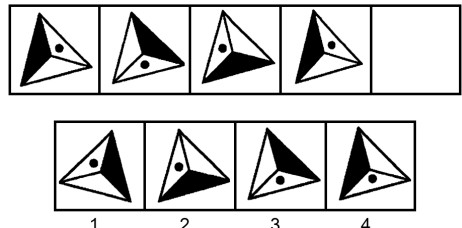

Figure 7: The example is formatted in *Sequence* configuration with the *Temporal Movement* pattern. The answer to this puzzle is C.

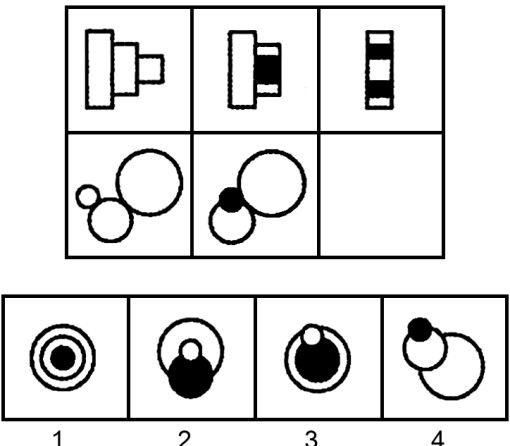

Figure 8: The example is formatted in *Two-row* configuration with the *Spatial Relationship* pattern. The answer to this puzzle is B.

## F Coarse-grained Perception on Different Categories

Table 6 shows MLLMs coarse-grained perception performance on different categories.

## G Experiment on Pattern Classification

To further understand the potential abstract visual reasoning abilities of MLLMs when perceptual barriers are mitigated, we replace the original AVR question with a pattern classification problem with the prompt shown in Table 13. We use regex mapping to extract MLLMs output, and the result visualized in Figure 5 is listed in Table 7

## H Experiment on Question with Text Description

To investigate how the models perform subsequent reasoning with accurate visual detail awareness, we randomly select 95 puzzles (five for each pattern in each task configuration) and provide text descriptions for each panel in the puzzle. To ensure a similar level of granularity for text descriptions, we first use GPT-4V to provide the original text descriptions, which human annotators further modify without introducing unrelated details (Figure 13).

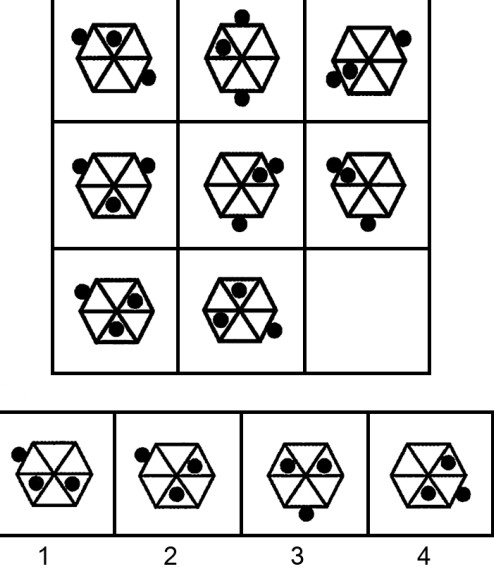

Figure 9: The example is formatted in *Matrix* configuration with the *Spatial Relationship* pattern. The answer to this puzzle is B.

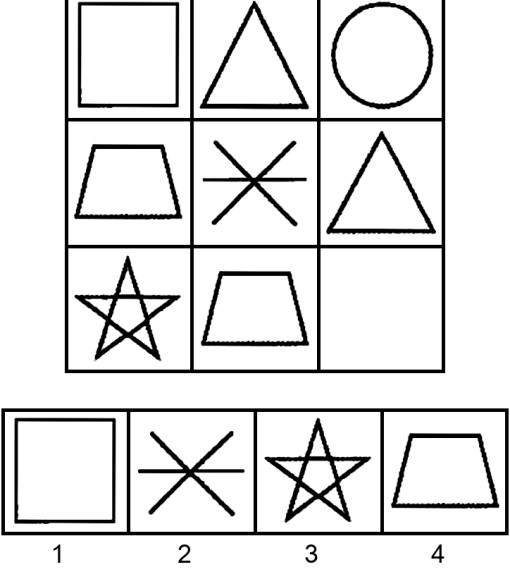

Figure 10: The example is formatted in *Matrix* configuration with the *Mathematical* pattern in *Geometric* shapes. The answer to this puzzle is A.

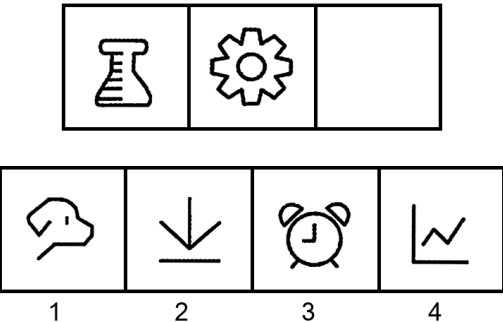

Figure 11: The example is formatted in *Group* configuration with the *2D-Geometry* pattern in *Abstract* input shapes. The answer to this puzzle is C.

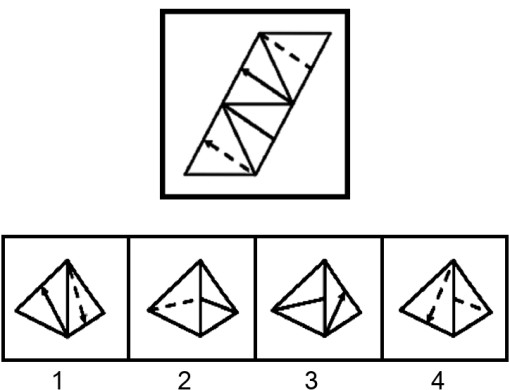

Figure 12: The example is formatted in *Reassembling* configuration with the *3D-Geometry* Quantities. The answer to this puzzle is B.

We input MLLMs with both the AVR question and text descriptions. The result is shown in Table 3. With the help of text descriptions, MLLMs, especially closed-sourced MLLMs, can build their abstract visual reasoning on correct visual detail foundations, gaining significant improvement in the performance (11.57% to 44.21%). GPT-4V even shows on-par performance (65.26%) with humans, highlighting the importance of visual perception ability. We also want to point out that some puzzles containing abstract shapes are challenging to describe. A tool that can convert images to SVGs and text descriptions [65] can be a possible approach to mitigate the difficulty and enhance MLLMs performance.

# I   Few-Shot COT

We show our prompting template in Table 13. We discuss few-shot performance in Tables 8 to 10 and 12. Table 8 shows few-shot result with Chain-of-Thought demonstrations in the prompt. Table 9 compares one-shot result with Chain-of-Thought demonstrations from same pattern or different pattern picked randomly (OOD). Table 10 compares one-shot result with Chain-of-Thought demonstrations from single task format (sequence) and two different task formats (sequence and two-row). Table 12 shows the model's few-shot performance on different patterns.

We also test whether breaking puzzles separately and prompting MLLMs with panels one by one can enhance abstract reasoning performance. We construct our prompt by adding *"This is the {index}*

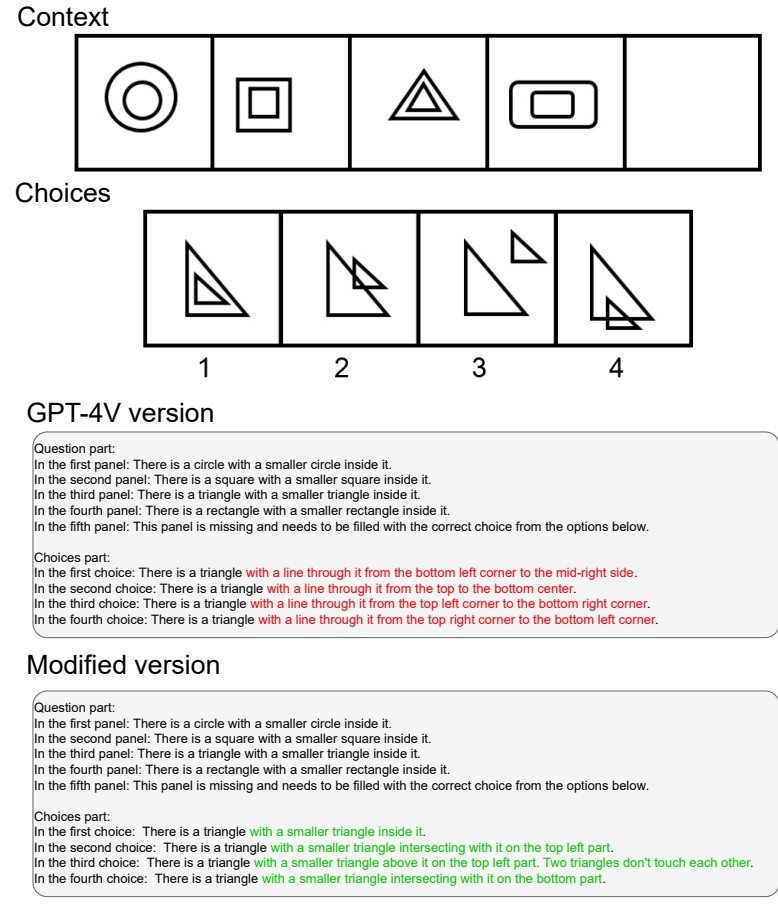

Figure 13: An example of the annotation process. We first generate text descriptions from GPT-4V, and the output will further be modified by human annotators (red → green).

Table 4: Performance of different models for different patterns. † notes the result is attributed to inductive bias.

| Pattern | Human | Open-sourced MLLMs | | | | | Closed-sourced MLLMs | | | | |
|---|---|---|---|---|---|---|---|---|---|---|---|
| | | Qwen-VL | Fuyu† | Blip-2† | InstructBLIP† | LLaVA-1.5 | GPT-4o | GPT-4V | Gemini† | Claude3 (Sonnet)† | Claude3 (Opus) |
| Temporal Movement | 82.08 | 23.81 | 24.76 | 25.71 | 25.71 | 20.95 | 23.81 | 26.67 | 22.86 | 23.81 | 31.43 |
| Spatial Relationship | 70.42 | 20.83 | 25.00 | 26.67 | 26.67 | 26.67 | 30.83 | 24.17 | 29.17 | 34.17 | 26.67 |
| Quantities | 81.57 | 15.76 | 24.85 | 27.88 | 27.27 | 28.48 | 29.09 | 21.21 | 24.85 | 24.85 | 27.88 |
| Mathematical | 60.00 | 21.25 | 25.00 | 24.17 | 24.17 | 28.75 | 26.25 | 21.67 | 23.75 | 23.33 | 27.50 |
| 2D-Geometry | 59.17 | 17.50 | 22.50 | 20.83 | 20.83 | 25.00 | 30.0 | 18.33 | 25.00 | 30.83 | 33.33 |
| 3D-Geometry | 50.00 | 15.00 | 15.00 | 15.00 | 15.00 | 15.00 | 25.00 | 30.00 | 30.00 | 20.00 | 25.00 |

*panel in the context of the puzzle"* to each panel in the context part and adding *"This is the {index} choice of the puzzle"* to each choice panel. Due to the limitation in resources, we sample 50 puzzles from MARVEL and compare zero-shot prompting settings with one-by-one prompting settings on Gemini and GPT-4V. As shown in Table 11, one-by-one prompting can not enhance model reasoning performance by providing panels separately. Thus, we decide to choose zero-shot prompting as our base prompting strategy.

## J   Human Evaluation

**AVR Question.** We provide each human annotator with the following instructions on AVR reasoning question evaluation:

Table 5: Performance of different models for different task configurations. † notes the result is attributed to inductive bias.

| Task format | Human | Open-sourced MLLMs | | | | | Closed-sourced MLLMs | | | | |
| --- | --- | --- | --- | --- | --- | --- | --- | --- | --- | --- | --- |
| | | Qwen-VL | Fuyu† | Blip-2† | InstructBLIP† | LLaVA-1.5 | GPT-4o | GPT-4V | Gemini† | Claude3 (Sonnet)† | Claude3 (Opus) |
| Sequence | 67.84 | 18.18 | 22.42 | 23.03 | 23.03 | 23.64 | 31.52 | 28.48 | 23.03 | 26.06 | 30.30 |
| Two-row | 71.56 | 16.00 | 22.67 | 24.89 | 24.44 | 25.78 | 25.78 | 22.22 | 24.00 | 25.78 | 32.00 |
| Matrix | 73.56 | 23.56 | 29.78 | 28.89 | 28.89 | 30.67 | 27.11 | 20.44 | 26.67 | 27.56 | 27.56 |
| Group | 58.18 | 21.48 | 21.48 | 21.48 | 21.48 | 25.19 | 28.15 | 17.04 | 25.93 | 27.41 | 24.44 |
| Reassembling | 50.00 | 15.00 | 15.00 | 15.00 | 15.00 | 15.00 | 25.00 | 30.00 | 30.00 | 20.00 | 25.00 |

Table 6: Coarse-grained perception performance on different categories bias.

| Category | Qwen-VL | Fuyu | Blip2 | InstructBLIP | LLaVA-1.5 | GPT-4o | GPT-4V | Gemini | Claude3 (Sonnet) | Claude3 (Opus) |
| --- | --- | --- | --- | --- | --- | --- | --- | --- | --- | --- |
| Location | 40.00 | 15.38 | 40.00 | 35.38 | 61.54 | 66.15 | 41.54 | 41.54 | 32.31 | 32.31 |
| Color | 37.74 | 26.42 | 58.49 | 58.49 | 58.49 | 69.01 | 52.83 | 54.72 | 47.17 | 62.26 |
| Shape | 33.33 | 36.16 | 45.20 | 36.72 | 29.38 | 69.08 | 45.76 | 53.67 | 57.63 | 45.76 |
| Quantity | 36.84 | 36.84 | 60.86 | 56.25 | 57.57 | 58.49 | 51.64 | 39.47 | 56.58 | 48.36 |
| Comparison | 40.94 | 40.35 | 52.05 | 53.22 | 57.31 | 70.62 | 60.82 | 41.52 | 42.11 | 47.95 |

*Welcome to our IQ Test Challenge! Test your cognitive skills and problem-solving abilities with a variety of questions in logical reasoning, pattern recognition, math, and more.*

*Remember, this test is a fun brain exercise, not a definitive measure of intelligence. There is no penalty for guessing. So, find a quiet spot, relax, and enjoy the challenge. Good luck! Instructions: There are five types of question formats in this test. Sequence, Two-row, Matrix, Group, and Reassembling.*

*Sequence: The question part on the top is a set of visual patterns arranged in a sequence. Find the pattern and add the missing pattern in the sequence from the choice below.*

*Two-row: The question part on the top contains two rows of images. Find the pattern in the first row and adapt the pattern on the second row.*

*Matrix: The question part is a set of visual patterns arranged in a 3 by 3 matrix, with the bottom right piece missing. The pattern can be found either along rows or columns.*

*Group: The question part on the top contains two images; one of the answers in the bottom shares the same pattern in the question images while other choices in the bottom part share a different pattern.*

*Reassembling: The question part on the top is an unfolded diagram. The answer part on the bottom contains 4 options that represent the correct three-dimensional assembly.*

**Fine-grained Perception Question.** We provide each human annotator with the same instructions shown in Table 13 on fine-grained perception question evaluation:

**Participant Demographics**. Our 30 human annotators range in age from 10 to 50 years. The group includes one elementary school student, three individuals with a bachelor's degree, 19 with a master's degree, and six PhD students. We released our human evaluation as a competition with current SOTA MLLMs. Participants in our human evaluation expressed enthusiasm for the test and provided valuable feedback. Most individuals adopted a strategy known as forward chaining [18], constructing evidence from the images and connecting it to the most likely candidate answer. One participant described their approach: "I identified some patterns to base my choice on. While they didn't always explain the full picture initially, they helped clarify certain aspects. I then iteratively identified more patterns to guide my final answer." Participants generally began by hypothesizing several possible patterns from the puzzle, then mapped these assumptions onto the available choices to determine the answer.

## Dataset Documentation and Intended Use

**Motivation** As multi-modal large language models (MLLMs) show promising progress in the visual reasoning domain, to what extent these models have abstract visual reasoning (AVR)abilities is still unknown. Also, the lack of a holistic AVR benchmark limited the current evaluation of these models, which motivates us to propose *MARVEL*, the first comprehensive multidimensional AVR benchmark. The dataset is created intentionally with the task in mind, aiming to evaluate MLLMs' abstract visual reasoning ability across different patterns, shapes and task configurations. The dataset was created by Yifan Jiang, Jiarui Zhang, Kexuan Sun, Zhivar Sourati, Kian Ahrabian, Kaixin Ma, Filip Ilievski and

Table 7: Performance of different models on pattern classification.

| Input | Open-sourced MLLMs | | | | | Closed-sourced MLLMs | | | | |
|---|---|---|---|---|---|---|---|---|---|---|
| | Qwen-VL | Fuyu | Blip-2 | InstructBLIP | LLaVA-1.5 | GPT-4o | GPT-4V | Gemini | Claude3 (Sonnet) | Claude3 (Opus) |
| AVR | 19.22 | 15.58 | 16.10 | 17.14 | 17.01 | 27.53 | 21.56 | 18.96 | 22.60 | 25.84 |

Table 8: Few-shot COT accuracy

| Model | zero-shot | one-shot | two-shot |
|---|---|---|---|
| Gemini | 25.06 | 28.7 | 26.75 |
| GPT-4o | 27.79 | 30.78 | 28.05 |
| GPT-4V | 22.34 | 26.1 | 28.31 |
| Claude3 (sonnet) | 26.49 | 25.97 | 26.23 |
| Claude3 (Opus) | 28.83 | 27.79 | 23.64 |

Jay Pujara from Information Sciences Institute, University of Southern California, Tencent AI Lab and Vrije Universiteit Amsterdam This research was sponsored by the Defense Advanced Research Projects Agency via Contract HR00112390061.

**Composition** The instances are abstract visual reasoning puzzle images, together with a set of AVR reasoning questions and perception questions to enable a hierarchical evaluation framework. There are 770 puzzles in total in the dataset. The puzzle is a sample of all possible instances. We root our dataset in human cognitive science to ensure the generality and applicability of our dataset. Each instance consists of a puzzle, an AVR reasoning question, a fine-grained perception question, as well as three coarse-grain questions. Each instance consists of a reasoning label representing the correct answer for AVR questions and a text answer for all perception questions. The whole dataset should be considered as an evaluation benchmark rather than a dataset supporting training, validation and testing. The dataset is entirely self-contained.

*Does the dataset contain data that might be considered confidential (e.g., data that is protected by legal privilege or by doctor-patient confidentiality, data that includes the content of individuals' non-public communications)?* No.

*Does the dataset contain data that, if viewed directly, might be offensive, insulting, threatening, or might otherwise cause anxiety?* No.

*Does the dataset identify any subpopulations (e.g., by age, gender)? If so, please describe how these subpopulations are identified and provide a description of their respective distributions within the dataset* No.

*Is it possible to identify individuals (i.e., one or more natural persons), either directly or indirectly (i.e., in combination with other data) from the dataset? If so, please describe how.* No.

*Does the dataset contain data that might be considered sensitive in any way (e.g., data that reveals race or ethnic origins, sexual orientations, religious beliefs, political opinions or union memberships, or locations; financial or health data; biometric or genetic data; forms of government identification, such as social security numbers; criminal history)?* No

**Collection Process** The dataset is collected from public available website using crawlers (mentioned in the main paper). All authors are involved in the data collection process. The instance is collected before Jan 2024.

*Did the individuals in question consent to the collection and use of their data?* No.

*If consent was obtained, were the consenting individuals provided with a mechanism to revoke their consent in the future or for certain uses? I* N/A.

*Has an analysis of the potential impact of the dataset and its use on data subjects (e.g., a data protection impact analysis) been conducted?* N/A.

**Preprocessing/cleaning/labeling** We filter the raw data after collecting data by removing duplicate, low-quality data manually. Three human annotators then choose the puzzle containing proper input

Table 9: Few-shot COT ablation with out-of-distribution (OOD) demonstration

| Model | one-shot | one-shot (OOD) |
|---|---|---|
| Gemini | 23.85 | 24.62 |
| GPT-4o | 28.08 | 23.08 |
| GPT-4V | 27.31 | 23.85 |
| Claude3 (sonnet) | 25.00 | 26.92 |
| Claude3 (Opus) | 29.62 | 25.00 |

Table 10: Few-shot COT ablation with mix distribution demonstration

| Model | two-shot | two-shot (mix) |
|---|---|---|
| Gemini | 30.48 | 24.29 |
| GPT-4o | 27.62 | 27.14 |
| GPT-4V | 29.52 | 28.10 |
| Claude3 (sonnet) | 28.57 | 23.81 |
| Claude3 (Opus) | 21.90 | 22.86 |

shapes and patterns in our predefined settings, which can further be re-organized in different task configurations.

*Was the "raw" data saved in addition to the preprocessed/cleaned/labeled data (e.g., to support unanticipated future uses)?* Yes.

*Is the software that was used to preprocess/clean/label the data available? If* Yes, the python package Pillow is used.

**Use** *Has the dataset been used for any tasks already?* No.

*What (other) tasks could the dataset be used for?* The task regarding analysing MLLMs' abstract visual reasoning abilities or visual perception abilities.

*Is there anything about the composition of the dataset or the way it was collected and preprocessed/cleaned/labeled that might impact future uses?* No.

*Are there tasks for which the dataset should not be used?* No.

**Distribution**

*Will the dataset be distributed to third parties outside of the entity (e.g., company, institution, organization) on behalf of which the dataset was created?* Yes. The dataset is available on the internet (`https://github.com/1171-jpg/MARVEL_AVR`).

*How will the dataset be distributed (e.g., tarball on website, API, GitHub)?* Yes. The link is presented in the last questions.

*When will the dataset be distributed?* The dataset was first released in April 2024.

*Will the dataset be distributed under a copyright or other intellectual property (IP) license, and/or under applicable terms of use (ToU)?*

*Have any third parties imposed IP-based or other restrictions on the data associated with the instances?* No.

*Do any export controls or other regulatory restrictions apply to the dataset or to individual instances? I* The dataset is intended for research use only and should not be used to make critical decisions about individuals' capabilities. Such misuse could cause undue pressure and anxiety for participants and may not accurately reflect their true potential or abilities in real-world scenarios.

**Maintenance**

*Who will be supporting/hosting/maintaining the dataset?* All the authors: Yifan Jiang, Jiarui Zhang, Kexuan Sun, Zhivar Sourati, Kian Ahrabian, Kaixin Ma, Filip Ilievski and Jay Pujara

Table 11: Compare zero-shot prompting with one-by-one prompting

| Model | zero-shot | one-by-one |
|-------|-----------|------------|
| Gemini | 20.00 | 13.33 |
| GPT-4V | 20.00 | 17.78 |

Table 12: Few-shot performance on different patterns

| | one -shot | | | | | two -shot | | | |
|---|---|---|---|---|---|---|---|---|---|
| | Gemini | GPT-4o | GPT-4V | Claude3 (Sonnet) | Claude3 (Opus) | Gemini | GPT-4V | Claude3 (Sonnet) | Claude3 (Opus) | |
| Temporal Movement | 24.76 | 28.57 | 27.62 | 17.14 | 29.52 | 33.33 | 17.14 | 28.57 | 24.76 | 18.10 |
| Spatial Relationship | 30.00 | 30.83 | 33.33 | 24.17 | 26.67 | 25.83 | 32.5 | 22.50 | 30.83 | 19.17 |
| Quantities | 21.21 | 25.42 | 27.88 | 27.88 | 26.67 | 25.45 | 24.17 | 31.52 | 25.45 | 27.27 |
| Mathematical | 27.92 | 30.3 | 28.75 | 25.42 | 26.67 | 25.83 | 30.91 | 27.50 | 22.92 | 22.08 |
| 2D-Geometry | 26.67 | 45.0 | 25.00 | 30.83 | 30.83 | 26.67 | 38.33 | 30.00 | 28.33 | 31.67 |
| 3D-Geometry | 25.00 | 25.0 | 35.00 | 45.00 | 30.00 | 20.00 | 20.00 | 35.00 | 40.00 | 20.00 |

*How can the owner/curator/manager of the dataset be contacted (e.g., email address)?* The contact emails are yjiang44@usc.edu, jzhang37@usc.edu, kexuansu@usc.edu

*Is there an erratum?* N/A.

*Will the dataset be updated (e.g., to correct labeling errors, add new instances, delete instances)* The update will be posted on the GitHub link (`https://github.com/1171-jpg/MARVEL_AVR`) and website (https://marvel770.github.io/).

*Will older versions of the dataset continue to be supported/hosted/maintained?* Yes, the update will be released using a different version number.

*If others want to extend/augment/build on/contribute to the dataset, is there a mechanism for them to do so?* Others may do so and should contact the original authors about incorporating fixes/extensions.

**Author Statement** We bear all responsibility in case of violation of rights, and we maintain the dataset for the long term to ensure it is accessible and organized.

**Croissant metadata and Licenese** The data is also released in hugging face (`https://huggingface.co/datasets/kianasun/MARVEL`) under apache-2.0 licenes. The Croissant metadata can be viewed and downloaded via `https://github.com/1171-jpg/MARVEL_AVR/blob/main/MARVEL_Croissant.json`


Table 13: Prompt examples for our experiments.

| Experiment | Prompt Example |
| --- | --- |
| AVR Question (Reassembling) | *[IMG]* You are given a puzzle. The puzzle consists of a context part on the top and the choices in the bottom. The context part on the top is an unfolded diagram of the 3D shape. The choices part on the bottom contains 4 options (marked by 1, 2, 3, or 4) represents the correct three-dimensional assembly. Which option (either 1, 2, 3, or 4) is the most appropriate answer? |
| AVR Question (Group) | *[IMG]* You are given a puzzle. The puzzle consists of a context part on the top and the choices at the bottom. The context part on the top is a set of visual patterns arranged in a three-by-one sequence, with the last piece missing. The choices part on the bottom contains four options (marked by 1, 2, 3, or 4). Which option (1, 2, 3, or 4) is the most appropriate answer to fill the missing part? |
| AVR Question (Matrix) | *[IMG]* You are given a puzzle. The puzzle consists of a context part on the top and the choices at the bottom. The context part on the top is a set of visual patterns arranged in a three by three matrix, with the bottom right piece missing. The choices part on the bottom contains four options (marked by 1, 2, 3, or 4). Which option (1, 2, 3, or 4) is the most appropriate answer to fill the missing part? |
| AVR Question (Sequence) | *[IMG]* You are given a puzzle. The puzzle consists of a context part on the top and the choices part on the bottom. The context part on the top is a set of visual patterns arranged sequentially, with the last piece missing. The choices part on the bottom contains four options (marked by 1, 2, 3, or 4). Which option (1, 2, 3, or 4) is the most appropriate answer to fill the missing part? |
| AVR Question (Two-row) | *[IMG]* You are given a puzzle. The puzzle consists of a context part on the top and the choices part on the bottom. The context part on the top is a set of visual patterns arranged in a two-by-three matrix, with the bottom right piece missing. The choices part on the bottom contains four options (marked by 1, 2, 3, or 4). Which option (1, 2, 3, or 4) is the most appropriate answer to fill the missing part? |
| Perception Question (fine-grained) | *[IMG]* You are given a puzzle. The puzzle consists of a context part on the top and the choices part on the bottom. The context part on the top is a set of visual grids arranged in a *m by n* sequence, with the last piece missing. The choices part on the bottom contains four choices (marked by 1, 2, 3, or 4). *[question]* |
| Perception Question (coarse-grained) | *[IMG]* You are given a puzzle. The puzzle consists of a context part on the top and the choices part on the bottom. The context part on the top contains some grids, with the last missing blank grid to be completed. The choices part on the bottom contains a sequence of grids representing the possible choices. How many grids, including the blank grid, are in the context part? |
| Perception Question (coarse-grained) | *[IMG]* You are given a puzzle. The puzzle consists of a question part on the top and the choices part on the bottom. The question part on the top contains some grids, with the last missing blank grid to be completed. The choices part on the bottom contains a sequence of grids representing the possible choices. How many grids are there in the choices part? |
| Perception Question (coarse-grained) | *[IMG]* You are given a puzzle. The puzzle consists of a question part on the top and the choices part on the bottom. The question part on the top contains some grids, with the last missing blank grid to be completed. The choices part on the bottom contains a sequence of grids representing the possible choices. How many grids, including the blank grid, are there in the whole puzzle? |
| Pattern Classification | You are given a puzzle. The puzzle consists of a question part on the top and the choices part in the bottom. The question part on the top is a set of visual panels arranged in a 1 by 5 sequence, with the last piece missing. Choices part on the bottom contains 4 choices (marked by 1, 2, 3, or 4). The puzzle contains one of the following patterns:

Temporal Movement patterns focus on the related position change or movement.
Spatial Relationship patterns focus on the objects' relative positional relationship.
Quantity patterns focus on the accuracy of number comprehension.
Mathematical patterns focus on elementary mathematical operations.
2D-Geometry patterns focus on the 2D-geometric shape and its properties.
3D-Geometry patterns focus on the 3D-geometric shape and its properties.

Please answer which pattern the puzzle is mostly related to. Output your final answer with the pattern name strictly. |

    (d) Did you include the total amount of compute and the type of resources used (e.g., type of GPUs, internal cluster, or cloud provider)? [N/A]

4. If you are using existing assets (e.g., code, data, models) or curating/releasing new assets...

    (a) If your work uses existing assets, did you cite the creators? [N/A]

    (b) Did you mention the license of the assets? [Yes] In Appendix.

    (c) Did you include any new assets either in the supplemental material or as a URL? [Yes] The new asset can be found via URL in the abstract and appendix.

    (d) Did you discuss whether and how consent was obtained from people whose data you're using/curating? [N/A] Public available data.

(e) Did you discuss whether the data you are using/curating contains personally identifiable information or offensive content? [Yes] In Appendix.

5. If you used crowdsourcing or conducted research with human subjects...

   (a) Did you include the full text of instructions given to participants and screenshots, if applicable? [Yes] In Appendix.

   (b) Did you describe any potential participant risks, with links to Institutional Review Board (IRB) approvals, if applicable? [N/A]

   (c) Did you include the estimated hourly wage paid to participants and the total amount spent on participant compensation? [N/A]

