# OpenReview forum: "MARVEL: Multidimensional Abstraction and Reasoning through Visual Evaluation and Learning"
_NeurIPS.cc/2024/Datasets_and_Benchmarks_Track — NeurIPS 2024 Track Datasets and Benchmarks Poster_

### Official Review · Reviewer_haaJ · 2024-06-23
**Paper 1595 Official Review**

**Rating:** 6
**Confidence:** 3

**Review:**

Please check all the sections below for detailed reviews.

**Strengths:**

- The paper presents a well-designed benchmark, MARVEL, that addresses the limitations of existing AVR benchmarks. The inclusion of diverse reasoning patterns, hierarchical evaluation framework, and perception questions ensures a comprehensive assessment of MLLMs' abstract visual reasoning abilities.
- The study tackles an important question regarding the abstract visual reasoning abilities of MLLMs. By highlighting the significant performance gaps compared to humans, the paper emphasizes the need for further research in this area and the limitations of current models.
- The paper is well-written and effectively communicates its objectives, methodology, and findings. The clarity of the writing enhances the readability and understanding of the research, making it accessible to a broad audience.
- The comprehensive experiments on ten representative MLLMs provide valuable insights into their performance on MARVEL. The analysis of perception questions uncovers the models' weaknesses in comprehending visual features, contributing to a deeper understanding of the connection between perception and reasoning abilities.

**Additional Feedback:**

None.

**Clarity:**

The paper exhibits overall good writing quality, and the figures and tables are well-designed and presented clearly. Additionally, the extensive appendix provides additional details that are not covered in the main paper.

**Correctness:**

The claims made in this paper are generally accurate. The benchmark is appropriately constructed, and the subsequent evaluations are generally robust, covering a wide range of language models.

**Documentation:**

The dataset is publicly released, and it is well-documented.

**Ethics:**

The paper is generally ethically justifiable, but it is better to also address the topic of salary or how the human annotators are compensated in the context of human annotation.

**Limitations:**

The authors have discussed two primary limitations of the benchmark. First, the paper acknowledges that ongoing research in human cognition may impact the understanding of innate human core knowledge, necessitating the continual updating of MARVEL's scope. Additionally, future research should explore alternative evaluation metrics to incorporate the fourth core knowledge related to goal-directed and interactive actions.

**Opportunities For Improvement:**

Generally, I found this paper interesting and quite sound. Some minor issues from my perspective are:

- I believe this dataset is indeed challenging for MLLMs, primarily because the task differs significantly from their pre-training corpus. Consequently, it is understandable that they perform poorly since they were never trained for such tasks. In this regard, I think conducting additional fine-tuning experiments could make the paper more comprehensive. Considering the relatively small size of the benchmark, the authors could try dividing it into 4:1 training and evaluation sets. They can then explore whether fine-tuning an MLLM, such as LLaVa, significantly improves performance.
- Importantly, AVR questions are also challenging for humans. This further highlights the difficulty of the task and raises concerns about the reliability of the evaluation benchmark. It might be beneficial to conduct expert evaluations to confirm the correctness of the benchmark.
- I cannot understand why the example's correct answer is D in the main figure... I believe the benchmark is even challenging to human and such performance (near random guess) makes sense.
- Another minor issue is the lack of analyses or discussions on how we can potentially enhance MLLMs' abstract visual reasoning capabilities. Including such insights would provide more valuable contributions to the paper.

**Relation To Prior Work:**

The authors have discussed their relation to prior works in a dedicated section, and the comparisons made are comprehensive.

**Summary And Contributions:**

The paper introduces a novel benchmark called MARVEL that evaluates the abstract visual reasoning (AVR) abilities of multi-modal large language models (MLLMs).

The benchmark addresses the limitations of existing AVR benchmarks by encompassing diverse reasoning patterns, including core knowledge patterns, geometric and abstract shapes, and various task configurations.

The authors curated a dataset of 770 high-quality puzzles from publicly available websites and provided a hierarchical evaluation framework that combines AVR questions with perception questions to assess model performance.

Comprehensive experiments on ten representative MLLMs revealed that these models showed near-random performance on MARVEL, with significant performance gaps compared to humans. The analysis of perception questions highlighted the models' struggle to comprehend visual features.

The contributions of the paper include the MARVEL benchmark, the evaluation framework, and insights into the connection between perception and reasoning abilities of MLLMs.

---

> ### Author Rebuttal · Authors · 2024-08-16
>
> Thank you for the constructive feedback and for investing the time to explore the content and context of our paper! Here is a list of answers together with a quick pointer to the place in the paper:
>
> **Additional Training Result**:
>
> Based on your suggestion, we conducted additional training. Specifically, we split the data into a 4:1 ratio for training and validation and fine-tuned an LLaVA-1.5 model using a smaller but more recent and advanced language model, Qwen-1.5B, for 10 epochs. The model was trained on both Marvel AVR questions (main task) and perception questions, with performance evaluated on the validation set after each epoch. For the main task, our evaluation results show an average accuracy of 19.76%, a maximum accuracy of 29.87%, and a minimum accuracy of 12.99% (with random chance being 25%). On the perception task, the evaluation results indicate an average accuracy of 59.74%, a maximum accuracy of 62.98%, and a minimum accuracy of 53.89% (with random chance being 50%). The result indicates that using simple finetuning for domain adaptation is challenging. Our findings are consistent with recent studies [1,2], which also observed that vision models struggle to achieve significant performance on perceptual vision tasks. The generally poor performance is not solely due to domain shifts. As noted by the authors of [1,2], the perception task is inherently complex and may require the integration of multiple tasks. While exploring potential improvement strategies is beyond the scope of this dataset paper, our perception question set can serve as a valuable evaluation benchmark for such efforts. We will include the experimental results in the final version of the paper to make it more comprehensive.
>
>
>
> **Correctness of the benchmark**:
>
> Our puzzles are sourced from a public test website and, in some cases, are used in selection exams, where accuracy is essential for fair evaluation. Three human annotators filtered and evaluated the puzzles based on reference answers to ensure correctness. Furthermore, evaluation results from 30 human participants showed a reasonable distribution, with 65% of participants scoring within 5 points of the average. We plan to provide more detailed information about the data quality in the appendix.
>
> **Illustration of Main Figure Example**:
>
> In the provided example, the pattern is as follows: in each panel, the number of elements on the left (either black stars or circles) increases by one. On the right side, the number of elements (white stars or circles) in the first panel equals the sum of the elements in the second and third panels. This example is particularly challenging, with only 28% of our human participants answering it correctly. Additionally, we manually wrote explanations for each puzzle in our dataset to ensure clarity. We will include these details in the camera-ready version and explain them in the paper.
>
> **Possible Improvement**:
>
> We are the first to consider visual perception a crucial intermediate stage in abstract visual reasoning (L 49-51). Our additional experiments, conducted within our evaluation framework, highlight the importance of visual perception and reasoning (Table and Sec 7) and their connection. In contrast, previous AVR benchmarks approached problem-solving as an end-to-end process (Lines 47-49), leading to highly overfitted models that struggle to generalize effectively. Our findings on the importance of perception correct this flawed training strategy, aligning with recent research [1,2]. As discussed in our response to the first question, simple fine-tuning is inadequate to address MLLMs' limitations in visual perception, and we suggest possible improvement direction on visual perception (L 299-316 and Appendix F). Developing effective solutions for this challenge lies beyond the scope of this dataset paper.
>
> **Payment of Human Annotators**:
>
> We released our human evaluation as a competition with current SOTA MLLMs. People are very interested and volunteer to be involved. We filtered 30 people based on their ages and education levels, and we will provide statistics in the appendix of the final version.
>
>
> **Reference For Rebuttal**
>
> [1] [Mathvista: Evaluating mathematical reasoning of foundation models in visual contexts.](https://arxiv.org/pdf/2310.02255) (Lu, Pan, et al., ICLR 2024).
>
> [2] [Vision language models are blind.](https://arxiv.org/abs/2407.06581)  (Rahmanzadehgervi, Pooyan, et al., Arxiv 2024).

---

### Official Review · Reviewer_mxzQ · 2024-07-26

**Rating:** 7
**Confidence:** 4
**Correctness:** The claims made in the submission loo…

**Review:**

Overall, this paper makes clear contributions to the field of machine learning for abstract visual reasoning tasks. I appreciate the fact that the dataset was mostly manually collected from online websites (instead of being synthetically generated such as RAVEN).

The paper writing can still be improved, especially by including more examples, and more clear definition of certain concepts.

**Strengths:**

1. MARVEL captures a wide range of reasoning patterns and task configurations rooted in core human cognitive knowledge, providing a thorough evaluation of MLLMs' abstract visual reasoning abilities.
2. The inclusion of both AVR and perception questions in a hierarchical framework allows for precise analysis of models' performance, distinguishing between perception and reasoning capabilities.
3. The study conducts extensive experiments on ten state-of-the-art MLLMs, revealing significant insights into their limitations.

**Additional Feedback:**

N/A

**Clarity:**

See my Opportunities For Improvement comments where I have listed clarity questions.

**Documentation:**

The dataset contains some documentations.

**Ethics:**

No.

**Limitations:**

See my opportunities for improvement comments. The dataset can benefit from more fine-grained annotations of, for example, shapes and locations of objects.

**Opportunities For Improvement:**

I think this paper can benefit from a revision to make certain concepts more clear, and also more examples to illustrate the dataset itself.

To name a few:
L124: I still don't understand what is reassembling format after reading this sentence multiple times. The authors should include examples for this.
L129: What's exactly the definition of geometric shapes and abstract shapes? For "geometric shapes," do you have annotations for the exact shape (e.g., polygon, circle, etc.) From figure 1, the only feature I noticed was that geometric shapes contain mostly straight lines and circles while abstract shapes contain other forms of curves. If that's the only difference, why do we have to make a difference?
L146: I don't understand why the authors need to discuss object core knowledge here. I don't think there is any "object-centric" annotations in the dataset, right? This is in contrast to, for example, RAVEN.
L179: What's a choice index and why do we need to remove them for perception questions...?
L187: What's "inductive bias"? Is this the right word to be used here?

Experiments:

Overall the experiments are very comprehensive. But I think the paper can be further improved if they can discuss the following question:

Are there any conclusions that we can draw from using the proposed dataset, but can not draw from existing datasets? For example, if we evaluate the models on RAVEN, can we still observe similar patterns? This can be justified by evaluating the same models on existing datasets. If so, why do we need yet another dataset? I would guess this new dataset contains patterns not seen in previous datasets, such as 3D shapes, etc. Does the inclusion of it provide new insights into MLLMs?

**Relation To Prior Work:**

The paper presented discussions, but comparisons on "what new conclusions we can draw" would be beneficial.

**Summary And Contributions:**

The paper introduces MARVEL, a benchmark for evaluating abstract visual reasoning (AVR) in multi-modal large language models (MLLMs). MARVEL includes 770 puzzles based on six core cognitive patterns, geometric and abstract shapes, and five task configurations. It features a hierarchical evaluation framework with both AVR and perception questions to assess models' reasoning and perception abilities. Experiments show that all tested MLLMs perform significantly worse than humans on MARVEL

MARVEL is one of the most comprehensive and human-created datasets for AVR evaluations.

---

> ### Author Rebuttal · Authors · 2024-08-16
>
> We genuinely appreciate your praise of our work and, even more, the comments and insightful suggestions! Here we list the answers to your comments:
>
> **Clarification of Reassembling Format**:
>
> The Reassembling format visualizes 3D geometric shapes, such as cubes or tetrahedrons, in the choice section of the puzzle. The context section provides a panel featuring the unfolded 2D diagram of one of the choices. Models must reason about the visual details in the 2D diagram to identify the correct 3D shape. An example involving a tetrahedron is provided in the appendix (Figure 12). To avoid confusion, we will elaborate on this description and include an example in the main content.
>
> **Definition of geometric shapes and abstract shapes:**
>
> Our definition of geometric and abstract shapes is based on the taxonomy outlined in a previous AVR review paper (L 127-128) [1]. Geometric shapes typically consist of fixed, easily describable basic forms such as polygons and circles, characterized by specific properties and generally belonging to a finite set of shapes. For example, both a larger square and a black square can be categorized as squares because they share fundamental properties—specifically, a square is defined as a shape with four equal sides and four right angles (L 129-130). In contrast, abstract shapes are atypical and do not adhere to fixed properties characterized by their variability and complexity. They rarely repeat or share the same properties across the dataset (L 134-135). Most previous AVR benchmarks first defined a set of geometric shapes and generated puzzles by modifying certain properties (e.g., size, color). For instance, RAVEN defines five geometric shapes: triangles, squares, pentagons, hexagons, and circles, and creates various shape variants by changing their size, rotation, and colors. While this approach can easily expand dataset size, it often leads to models excelling in evaluations through superficial learning, resulting in biased evaluation. Abstract shapes are preferred in AVR research, according to [1], as they allow for a more effective assessment of a model's true reasoning capabilities, reducing the risk of overfitting to specific geometric norms (L 135-136).  We are the first benchmark to incorporate both geometric and abstract shapes to ensure a comprehensive evaluation (Table 1), and we have a geometric set that includes shapes from previous AVR benchmarks (Table 1). Each puzzle has been annotated for the presence of geometric and abstract shapes, with 38.8% of puzzles containing abstract shapes and 87.4% containing geometric shapes. We further have a detailed explanation of each puzzle in our dataset link. We will provide additional illustrations in the camera-ready version of the main content.
>
> **Usage of Object Core Knowledge**:
>
> To ensure our MARVEL pattern captures real-world complexity, we adapt Core knowledge theory from cognitive science as our theoretical foundations (L 138-145). Object Core Knowledge belongs to one of the Core knowledge theories, and we develop two abstract reasoning patterns, the *Temporal Movement Pattern*, and the *Spatial Relationship Pattern*, based on that (L 146-149). It is a concept related to object boundary and movement in cognitive science and has no relation with actual object annotations.
>
> **Clarification on the index**:
>
>  The choice index is the number marking each choice panel (1,2,3 and 4). We remove the choice index while evaluating the perception because coase-grained questions ask the number of panels in context, choice and the whole puzzle (L 177-179). The number in the choice index can provide a hint, and we removed it to avoid short-cut learning. We will emphasize this point in the final version.
>
> **Clarification on the inductive bias**:
>
> The inductive bias (also known as learning bias) of a learning algorithm is the set of assumptions that the learner uses to predict outputs of given inputs that it has not encountered. In our paper, for example, we prevent the model from obtaining a high performance just because it always outputs 'D' and the correct answer is coincidentally mostly 'D'. We illustrate the inductive bias we found in Lines 231 to 240. We will add examples for illustration in the place you point out.

---

> > ### Author Rebuttal · Authors · 2024-08-16
> >
> > **New insight from MARVEL**:
> >
> > MARVEL offers significant new insights as follows:
> > 1) **Visual Perception and Reasoning**: Our experimental results reveal that MLLMs exhibit weak visual perception abilities (Table 2) and a gap in visual reasoning performance between open-source and closed-source MLLMs, even after providing accurate text descriptions of puzzles (Table 3 and Figure 5). Additional results in the appendix show that while MLLMs perform well in color perception, they struggle with location recognition (L 310-312 and Appendix F). MARVEL emphasizes the importance of visual perception—an aspect often overlooked by previous AVR benchmarks—by separately evaluating visual perception and reasoning abilities, making the evaluation more precise and diagnosable (L 47-49).
> >
> > 2) **Performance on 2D-Geometry Patterns**: Current MLLMs achieve their best performance on the 2D-Geometry pattern compared to other patterns (L 290-292). MARVEL introduces a wider variety of reasoning patterns (detailed in Table 1), allowing for a more comprehensive assessment of MLLM capabilities and enabling a detailed analysis of model performance across different patterns. Previous benchmarks often combined multiple patterns within a single puzzle or lacked clear annotations for each pattern.
> >
> > 3) **Challenges with 3D-Geometry Patterns**: The newly designed 3D-Geometry pattern, grounded in cognitive science theory, presents a significant challenge for current MLLMs (L 287-289). MARVEL develops reasoning patterns grounded in cognitive science theory, ensuring thorough coverage. The newly designed 3D-Geometry pattern, which is critical for practical applications (Lines 36-37), is included, addressing a gap left by previous benchmarks that lacked a theoretical foundation.
> >
> > 4) **Bias in Diverse Configurations**: Our analysis of results based on task configuration highlights the potential bias introduced by single-configuration evaluations (L 296-297). By incorporating a variety of task configurations and including both geometric and abstract shapes, MARVEL ensures a more robust evaluation. Prior research has shown that evaluations based on datasets with a single task configuration and only geometric shapes may introduce biases toward specific settings [1].
> >
> >
> > **Reference For Rebuttal**
> >
> > [1] [A review of emerging research directions in abstract visual reasoning.](https://www.sciencedirect.com/science/article/abs/pii/S1566253522002214) (Małkiński, Mikołaj, and Jacek Mańdziuk., Information Fusion 2023)

---

> > > ### Comment · Reviewer_mxzQ · 2024-08-29
> > >
> > > I want to thank the authors for answering my questions and addressing my concerns.
> > >
> > > I am satisfied with most of the responses except for the one on "core knowledge." I am not sure if the patterns are really connected to cognitive science experiments (e.g., can they be used in human subject experiments?) But I think this is fine because this dataset is not primarily designed for cognitive science studies.
> > >
> > > In their revision, the authors should definitely include more examples and these clarifications (for example, inductive bias is usually associated with the prior of the learning algorithm and the model class, such as the usage of a CNN over a fully-connected MLP, but not biases in the data. It's fine to use it this way, but the authors should provide explanations).
> > >
> > > I maintain my score and believe that this is a valuable contribution.

---

> > > > ### Author Rebuttal · Authors · 2024-08-29
> > > >
> > > > We appreciate the reviewer's meaningful suggestion. We will include more details, such as more examples and explanations, in the camera-ready version.

---

### Official Review · Reviewer_PeKX · 2024-08-07
**A Promising but Potentially Limited Multi-Dimensional Abstract Visual Reasoning Benchmark for MLLMs**

**Rating:** 7
**Confidence:** 3
**Correctness:** Correct.

**Review:**

This work demonstrates high quality and originality in several aspects. The design of the MARVEL benchmark, covering multiple patterns and task configurations, provides a solid foundation for comprehensive evaluation of MLLMs. The introduction of the hierarchical evaluation framework, particularly the combination of perception questions with AVR questions, enhances the interpretability of results. The experimental design is rigorous, including evaluation of both open-source and closed-source models, as well as testing in zero-shot and few-shot settings.

However, the work also has some limitations. The main concern is whether the benchmark truly reflects MLLMs' abstract visual reasoning ability, as results suggest the bottleneck may be visual perception rather than reasoning. The small dataset size may lead to unstable evaluation and overfitting. Additionally, the relationship between MARVEL and existing benchmarks, as well as the value of improving MLLMs' abstract visual reasoning capabilities, needs further clarification.

**Pros:**
- Comprehensive coverage of multiple patterns and task configurations
- Improved hierarchical evaluation framework
- Rigorous experimental design
- Detailed documentation for reproducibility

**Cons:**
- Potential misalignment between benchmark design and intended measurement
- Limited dataset size
- Unclear positioning relative to existing benchmarks
- Insufficient justification for the value of improving MLLMs' AVR abilities

**Strengths:**

- **Comprehensive Coverage:** MARVEL covers multiple patterns (six core knowledge patterns) and task configurations (five distinct types), allowing for a more thorough assessment of MLLMs' abilities across various abstract reasoning scenarios.

- **Improved Evaluation Approach:** The evaluation approach is more reasonable and interpretable compared to prior works. The introduction of perception questions alongside AVR questions allows for a more nuanced understanding of model performance, helping to isolate whether errors stem from perception issues or reasoning failures.

- **Rigorous Experimental Design:** The experimental design is thorough in several aspects:
  1. Evaluates both open-source and closed-source MLLMs
  2. Includes zero-shot and few-shot settings to test model adaptability
  3. Establishes a human baseline for performance comparison

- **Reproducibility and Accessibility:** The paper facilitates easy use and reproduction by:
  1. Providing detailed descriptions of data collection and curation processes
  2. Clearly explaining the rationale behind MARVEL's design choices
  3. Releasing the dataset for public use

**Additional Feedback:**

Overall, the **Benchmark Validity** issue mentioned above is my primary concern. Addressing this point would significantly support the value of this benchmark. Alternatively, if the authors could address the remaining issues of Dataset Size, Benchmark Positioning, and Value Justification, it would also enhance my recognition of this work's value as an improvement over previous efforts.

**Clarity:**

Well written. It would be better if the authors could provide some examples of human's "Chain of Thoughts" when solving those puzzles.

**Documentation:**

Yes.

**Ethics:**

No.

**Limitations:**

See limitations and suggestions above.

**Opportunities For Improvement:**

- **Benchmark Validity:** My main concern is whether this benchmark truly reflects the abstract visual reasoning ability of MLLMs. The authors' inclusion of a visual perception component is a good point, but the results from this section seem to suggest that the bottleneck for MLLMs on this benchmark is not their reasoning ability, but simply their insufficient visual perception capabilities. Additionally, as the authors mention, when precise text descriptions are provided, the performance of various models improves significantly. The value of this benchmark would be greatly diminished if MLLMs perform poorly simply because they haven't encountered images in this domain before.  I personally suggest that the authors could:
  - Implement a visual perception difficulty curriculum by providing text descriptions of varying degrees
  - Present some results after simple domain adaptation of the visual perception componen

- **Dataset Size:** The dataset size is somewhat small, which may lead to unstable evaluation results. Particularly, even after MARVEL covers 2 Input Shapes, 6 Patterns, and 5 Configurations, it still only has 770 samples. Does this mean that there are very few questions to measure each item, which would make the evaluation very sensitive and prone to overfitting? (For comparison, RAVEN only covers 1 Input Shape, 3 Patterns, and 1 Configuration, yet has 70,000 questions.)

- **Benchmark Positioning:** The comparison and positioning with previous benchmarks are unclear. Compared to previous benchmarks, is MARVEL intended as a replacement or a supplement? Can the evaluation results from MARVEL bring more valuable insights compared to previous benchmarks?

- **Value Justification:** There is insufficient elaboration on the value of measuring and improving MLLMs' AVR abilities. In particular, the authors should motivate the consideration of more Input Shapes, Patterns, and Configurations in MARVEL using practical value or inspiration for subsequent research. Otherwise, it becomes meaningless and may make the evaluation less insightful.

**Relation To Prior Work:**

Basically clear. See limitations above for insufficient parts.

**Summary And Contributions:**

This paper introduces MARVEL, a novel multi-dimensional abstract visual reasoning (AVR) benchmark for evaluating multi-modal large language models (MLLMs). MARVEL features six core knowledge patterns across five task configurations, along with a hierarchical evaluation framework that includes both perception and reasoning questions. The authors tested several open-source and closed-source MLLMs using MARVEL, revealing significant gaps between current models and human performance in AVR tasks. Key contributions include the development of the benchmark itself, the design of the evaluation framework, and the comprehensive testing of state-of-the-art MLLMs. The study highlights the importance of visual perception in abstract reasoning and identifies major limitations in current MLLMs, particularly in their ability to accurately perceive and interpret visual details in abstract contexts.

---

> ### Author Rebuttal · Authors · 2024-08-16
>
> We appreciate your recognition of the value of our research topic! We really appreciate your thorough suggestions, and here we list the main responses in relation to your concerns:
>
> **Experiment on Domain Adaptation**
>
> In response to the reviewer’s suggestion, we conducted additional training. Specifically, we split the data into a 4:1 ratio for training and validation and fine-tuned an LLaVA-1.5 model using a smaller but more recent and advanced language model, Qwen-1.5B, for 10 epochs. The model was trained on Marvel AVR questions (main task) and perception questions, with performance evaluated on the validation set after each epoch. For the main task, our evaluation results show an average accuracy of 19.76%, a maximum accuracy of 29.87%, and a minimum accuracy of 12.99% (with random chance being 25%). On the perception task, the evaluation results indicate an average accuracy of 59.74%, a maximum accuracy of 62.98%, and a minimum accuracy of 53.89% (with random chance being 50%). The result indicates that using simple finetuning for domain adaptation is challenging. Our findings are consistent with recent studies [1,2], which also observed that vision models struggle to achieve significant performance on perceptual vision tasks. The generally poor performance is not solely due to domain shifts. As noted by the authors of [1,2], the perception task is inherently complex and may require the integration of multiple tasks. While exploring potential improvement strategies is beyond the scope of this dataset paper, our perception question set can serve as a valuable evaluation benchmark for such efforts. We will include the experimental results in the final version of the paper to make it more comprehensive.
>
> **Benchmark Validity**:
>
> As mentioned in Lines 49-51, visual perception is a fundamental component of the visual reasoning process. Our experiments across SOTA MLLMs on perception questions underscore their weak visual perception abilities (L 47-49). Additional experiments with text descriptions further confirm that visual perception can act as a bottleneck (Section 7). However, as shown in Figure 5 and Table 3, there is a clear performance gap between open and closed-source models, and this gap widens when accurate text descriptions are provided (Lines 352-354). This suggests that visual perception is not the only bottleneck for MLLMs. We also analyzed human performance on this subset, finding that human participants outperformed the models with a performance of 87.2%, while the best model (GPT-4) achieved only 65.3%. Additionally, based on your suggestion, we conducted a domain adaptation experiment (please refer to the **Experiment on Domain Adaptation**), and the results indicate that poor performance is not simply due to unfamiliar images and cannot be easily resolved through fine-tuning. In Sec 7, the two additional experiments we provide have different difficulty levels of text description.
>
> In summary, the main evaluation results (Table 2) and additional perception experiment (Sec 7) on MARVEL demonstrate that visual perception and reasoning are both critical bottlenecks for MLLMs when performing abstract visual reasoning. The text description results in Section 7 reveal differences in visual reasoning abilities among MLLMs, supporting the validity of MARVEL. As more research highlights the weak visual perception abilities of current SOTA MLLMs [1,2,7], there is an increasing need for perception-based benchmarks. The carefully designed perception questions in MARVEL are highly valuable in addressing this urgent gap in the literature. We will enrich the main content with fine-tuning experiments in the final version.
>
> **Data Size**:
>
> 1) **Evaluation-Only Benchmark**: MARVEL is designed solely as an evaluation benchmark. Other AVR benchmarks, such as ARC, provide only 200 data points across four patterns for private evaluation, while the recent RAVEN IQ test dataset[7], IQ50, contains just 50 samples. MathVista[1], a popular benchmark for mathematical reasoning, uses 1,000 examples across 12 types for its public leaderboard. In comparison, MARVEL offers a larger set of puzzles, averaging over 100 puzzles per pattern.
>
>  2) **Issues with Extensive Training Sets and Single Task Configuration**: The RAVEN dataset is commonly used for both training and testing, but recent studies [4,5] have criticized RAVEN-like datasets for their extensive training sets, raising concerns about the generalizability of reported improvements. Furthermore, RAVEN evaluates models using a single task configuration (L 42-44), leading to highly specialized models that may be overfitted to a specific problem [6].
>
> 3) **Evaluation Stability**: MARVEL is the first AVR benchmark to include multiple configurations for each pattern. When evaluating MLLMs on different AVR patterns, the results are averaged across all configurations, ensuring the validity of the outcomes and reducing the risk of overfitting (L 295-298). Additionally, all puzzles in MARVEL are either manually collected or handcrafted (Section 3.4), avoiding the short-learning issues often observed with synthetic datasets like RAVEN.

---

> > ### Author Rebuttal · Authors · 2024-08-16
> >
> > **Benchmark Positioning**:
> >
> > MARVEL is a supplement for previous benchmarks in the following points:
> >
> > 1) **Broader Range of Configurations, Shapes, and Input Types**: MARVEL incorporates patterns grounded in cognitive science (L 138-145) to cover a wide array of real-world scenarios. Notably, it introduces a 3D-Geometric pattern—an essential component for AVR-related practical applications, such as human pose estimation (L 35-37), which previous benchmarks have largely overlooked. By encompassing five distinct configurations and including both abstract and geometric shapes, MARVEL enables fairer comparisons between MLLMs, mitigating overfitting and sensitivity issues found in prior benchmarks (L 42-44). Our experiments across various task configurations also reveal biases inherent in previous single-configuration evaluations (L 296-297).
> >
> > 2) **Separation of Perception and Reasoning**: As outlined in the Benchmark Validity section, prior AVR benchmarks did not account for visual perception in their design (L 92-94). This oversight has led to improvement approaches focused on end-to-end training, resulting in brittle models that fail to generalize beyond the training tasks. MARVEL, in contrast, treats visual perception as a crucial intermediate stage in visual reasoning, demonstrating its importance through well-structured experiments (Section 7). Additional results in the appendix underscore that while MLLMs excel in color perception, they struggle with location recognition (L 310-312 and Appendix F).
> >
> > 3) **Perception-Focused Evaluation**: MARVEL provides fine-grained and coarse-grained perception questions, offering a unique platform for evaluating MLLMs' visual perception abilities within the AVR domain. Despite the growing recognition of perception in contemporary research [1,2,7], perception-based benchmarks remain scarce, and most evaluations depend on human assessments or case studies. The perception questions in MARVEL hold promise not only for advancing the AVR domain but also for applications in areas such as mathematical reasoning.  We plan to extract the perception questions from MARVEL and develop a perception-only benchmark to address the current literature gap. We will include this in the camera-ready version and move some relevant tables to the main content, where an additional content page will be provided.
> >
> > As mentioned in the **Data Size** section, the current size is sufficient for benchmarking.  We will also explore expanding it to enhance the supplementary material in future work.
> >
> > **Value Justification**:
> > 1) **Intelligence Approximation**. AVR problems often approximate human intelligence and have been recognized as a proxy for intelligence in previous literature (L 37-39). MARVEL’s inclusion of a wider range of input shapes, patterns, and configurations ensures a more robust and fair evaluation. By incorporating both abstract and geometric shapes, MARVEL mitigates the advantage that MLLMs might gain from exposure to similar geometric shapes in their training data (L 135-136). The diverse configurations also help prevent overfitting on single-task configurations, which is a common issue in previous benchmarks (L 295-297).
> >
> > 2) **Practical Application**. The AVR domain holds substantial potential for influencing various practical applications (L 35-39). For example, Relation Networks—one of the most widely adopted models based on AVR—have already been applied to tasks such as 3D human pose estimation [8], action recognition [9], and self-supervised learning of visual representations [10]. MARVEL incorporates patterns grounded in cognitive science (L 138-145) that capture a broader spectrum of real-world scenarios, addressing gaps overlooked by previous benchmarks (L 40-42). For instance, the 3D-Geometric pattern is essential for tasks like 3D human pose estimation, yet no prior AVR benchmark has included it. We will expand on these practical applications in the introduction and emphasize AVR’s role as an approximation of intelligence.
> >
> > **Human chain of thoughts examples**:
> >
> > Participants in our human evaluation expressed enthusiasm for the test and provided valuable feedback. Most individuals adopted a strategy known as forward chaining [3], constructing evidence from the images and connecting it to the most likely candidate answer. One participant described their approach: “I identified some patterns to base my choice on. While they didn’t always explain the full picture initially, they helped clarify certain aspects. I then iteratively identified more patterns to guide my final answer.” Participants generally began by hypothesizing several possible patterns from the puzzle, then mapped these assumptions onto the available choices to determine the answer. We will include human feedback and our conclusions in the appendix.

---

> > ### Author Rebuttal · Authors · 2024-08-16
> >
> > **Reference For Rebuttal**
> >
> > [1] [Mathvista: Evaluating mathematical reasoning of foundation models in visual contexts.](https://arxiv.org/pdf/2310.02255) (Lu, Pan, et al., ICLR 2024).
> >
> > [2] [Vision language models are blind.](https://arxiv.org/abs/2407.06581)  (Rahmanzadehgervi, Pooyan, et al., Arxiv 2024).
> >
> > [3] [Introduction to natural language processing.](https://www.google.com/books/edition/Introduction_to_Natural_Language_Process/72yuDwAAQBAJ?hl=en&gbpv=1&dq=Introduction+to+natural+language+processing&pg=PR5&printsec=frontcover)  (Eisenstein, Jacob, MIT press 2019)
> >
> > [4] [Abstraction and analogy‐making in artificial intelligence.](https://nyaspubs.onlinelibrary.wiley.com/doi/abs/10.1111/nyas.14619?casa_token=0iJZ4BqdBIcAAAAA%3AmSmR3W6tfUfCIoabkxSu52Y5pU4ujKO_r2wh71Uje6TVVh0KgrwT1T63dvAha6vKwUrZMRu-Wqxfqi7PFQ) (Mitchell, Melanie.,  Annals of the New York Academy of Sciences 2021)
> >
> > [5]  [Evaluating the progress of deep learning for visual relational concepts.](https://jov.arvojournals.org/article.aspx?articleid=2777974) (Stabinger, Sebastian, et al., Journal of Vision 2021)
> >
> > [6] [A review of emerging research directions in abstract visual reasoning.](https://www.sciencedirect.com/science/article/abs/pii/S1566253522002214) (Małkiński, Mikołaj, and Jacek Mańdziuk., Information Fusion 2023)
> >
> > [7] [Language is not all you need: Aligning perception with language models.](https://proceedings.neurips.cc/paper_files/paper/2023/hash/e425b75bac5742a008d643826428787c-Abstract-Conference.html) (Huang, Shaohan, et al., NeurIPS 2023)
> >
> > [8] [3d human pose estimation with relational networks.](https://arxiv.org/abs/1805.08961) (Park, Sungheon, and Nojun Kwak., Arxiv 2018).
> >
> > [9] [Actor-centric relation network.](https://openaccess.thecvf.com/content_ECCV_2018/html/Chen_Sun_Actor-centric_Relation_Network_ECCV_2018_paper.html) (Sun, Chen, et al., ECCV 2018)
> >
> > [10] [Self-supervised relational reasoning for representation learning.](https://proceedings.neurips.cc/paper_files/paper/2020/hash/29539ed932d32f1c56324cded92c07c2-Abstract.html)  (Patacchiola, Massimiliano, and Amos J. Storkey., NeurIPS 2020)

---

> > ### Comment · Reviewer_PeKX · 2024-08-18
> >
> > I appreciate the authors' rebuttal and detailed responses. Most of my concerns have been addressed. While I still have some concerns about the Benchmark Validity, I agree that resolving these issues may indeed be beyond the scope of this paper. Overall, I believe this work is a good improvement and expansion on previous benchmarks in the field, and I'd like to raise my score.
> >
> > To further enhance the clarity of this benchmark, I recommend that the authors incorporate the following elements into the manuscript:
> >
> > 1. **Integration of adaptation results**: The authors should include the adaptation results within the main body of the paper, particularly by presenting them alongside corresponding results in the same tables for direct comparison.
> >
> > 2. **Elaboration on AVR task practical significance**: It would be beneficial for the authors to provide a more thorough discussion on the practical value and real-world applications of the Audio-Visual Retrieval (AVR) task. This would help readers better understand its relevance and potential impact.
> >
> > 3. **Enhanced quantitative comparisons**: To strengthen the benchmark's positioning within the field, the authors should include more quantitative comparisons with other existing benchmarks. For instance, they could consider presenting ratios between numbers of samples and Input Shapes/Patterns/Configurations. This will provide a clearer picture of how their benchmark compares to others in terms of complexity and coverage.

---

> > > ### Author Rebuttal · Authors · 2024-08-18
> > >
> > > Thank you for your thoughtful feedback and for considering our rebuttal. We appreciate your recognition of our contributions and your suggestions for improvement. We will incorporate the recommended elements into the main content. These additions will further clarify the value and positioning of our work within the field. We sincerely appreciate your support and suggestions.

---

### Decision · Program_Chairs · 2024-09-26

**Decision:**

Accept (Poster)

**Comment:**

All reviewers were positive about the paper. They liked the multi-dimensional design and hierarchical evaluation of the abstractive visual reasoning benchmark, as well as the extensive experiments. Hence the decision is to recommend the paper for acceptance.

However, the reviewers also suggested new results and discussions, most of which the authors have addressed in the rebuttal. The authors are encouraged to incorporated them in the revision.